# Efficient generation of marmoset primordial germ cell-like cells using induced pluripotent stem cells

Yasunari Seita[1,2,3†], Keren Cheng[1,2†], John R McCarrey[4], Nomesh Yadu[4], Ian H Cheeseman[5,6], Alec Bagwell[5,6], Corinna N Ross[5,6], Isamar Santana Toro[4], Li-hua Yen[4], Sean Vargas[7], Christopher S Navara[4]*, Brian P Hermann[4,7]*, Kotaro Sasaki[1,2,8]*

[1]Department of Biomedical Sciences, University of Pennsylvania, School of Veterinary Medicine, Philadelphia, United States; [2]Institute for Regenerative Medicine, University of Pennsylvania, Philadelphia, United States; [3]Bell Research Center for Reproductive Health and Cancer, Nagoya, Japan; [4]Department of Neuroscience, Developmental and Regenerative Biology, The University of Texas at San Antonio, San Antonio, United States; [5]Texas Biomedical Research Institute, San Antonio, United States; [6]Southwest National Primate Research Center, San Antonio, United States; [7]Genomics Core, The University of Texas at San Antonio, San Antonio, United States; [8]Department of Pathology and Laboratory Medicine, University of Pennsylvania, Philadelphia, United States

*For correspondence:
christopher.navara@utsa.edu
(CSN);
Brian.Hermann@utsa.edu (BPH);
ksasaki@upenn.edu (KS)

†These authors contributed
equally to this work

Competing interest: The authors
declare that no competing
interests exist.

Reviewing Editor: Marianne E
Bronner, California Institute of
Technology, United States

**Abstract** Reconstitution of germ cell fate from pluripotent stem cells provides an opportunity to understand the molecular underpinnings of germ cell development. Here, we established robust methods for induced pluripotent stem cell (iPSC) culture in the common marmoset (*Callithrix jacchus* [cj]), allowing stable propagation in an undifferentiated state. Notably, iPSCs cultured on a feeder layer in the presence of a WNT signaling inhibitor upregulated genes related to ubiquitin-dependent protein catabolic processes and enter a permissive state that enables differentiation into primordial germ cell-like cells (PGCLCs) bearing immunophenotypic and transcriptomic similarities to pre-migratory cjPGCs in vivo. Induction of cjPGCLCs is accompanied by transient upregulation of mesodermal genes, culminating in the establishment of a primate-specific germline transcriptional network. Moreover, cjPGCLCs can be expanded in monolayer while retaining the germline state. Upon co-culture with mouse testicular somatic cells, these cells acquire an early prospermatogonia-like phenotype. Our findings provide a framework for understanding and reconstituting marmoset germ cell development in vitro, thus providing a comparative tool and foundation for a preclinical modeling of human in vitro gametogenesis.

## Editor's evaluation

This nicely done paper describes a method for robust differentiation of the common marmoset induced pluripotent stem cells (iPSCs) into primordial germ cell-like cells and subsequently into spermatogonia-like cells when combined with testis somatic cells. The data suggest that marmosets are very similar to humans and macaques.

## Introduction

The germline, a lineage that ultimately form the gametes, is the fundamental component of the life cycle in metazoan species, ensuring perpetuation and diversification of the genome across generations. In addition, the germline is the foundation of totipotency, since combination of gametes at fertilization gives rise to totipotent zygotes that establish all embryonic and extraembryonic lineages necessary for production of a new organism. The germline first arises during early embryonic development as primordial germ cells (PGCs), which subsequently migrate to the developing gonads and ultimately produce either spermatozoa or oocytes through complex and sex-specific developmental pathways (*Saitou and Miyauchi, 2016*). Accordingly, aberrancies associated with PGC development can lead to infertility and a variety of genetic and epigenetic disorders in offspring. Therefore, a precise understanding of how PGCs develop bears significant implications not only for reproductive medicine but also toward a better understanding of a breadth of human diseases.

Although much has been learned from murine genetic studies regarding the cellular dynamics, signaling, genetic, and epigenetic requirements accompanying PGC specification (*Saitou and Miyauchi, 2016*; *Saitou and Yamaji, 2012*), the scarcity of germ cells and complexity of their development and cellular interactions has limited deep understanding of transcriptional regulatory networks and epigenetic bases of germ cell development. The last decade, however, has witnessed remarkable progress toward establishing in vitro gametogenesis (IVG) technologies as an alternative approach to study germ cell development. Remarkably, through the stepwise recapitulation and validation of developmental milestones starting with pluripotent stem cells (embryonic stem cells [ESCs] or induced pluripotent stem cells [iPSCs]), the entirety of mouse germline development has been reconstituted in vitro, culminating in the successful generation of fertilization-competent oocytes and spermatozoa, and healthy offspring (*Hikabe et al., 2016*; *Ishikura et al., 2021*). These landmark studies have been followed by successful development of human iPSC-based germline reconstitution methods, in which pre-meiotic oogonia and prospermatogonia-like cells generated through PGC-like cells (PGCLCs) bear remarkable transcriptional similarities to in vivo counterparts (*Hwang et al., 2020*; *Yamashiro et al., 2018*; *Sasaki et al., 2015*).

IVG platforms have provided valuable tools to dissect the transcriptional and epigenetic mechanisms underlying germline specification and subsequent gametogenesis. Recent studies using IVG-derived germ cells or primate embryos in vivo have revealed a substantial divergence in the origin of germ cells and transcriptional networks governing germ cell specification between mice and humans (*Saitou and Miyauchi, 2016*). For example, in mice, core germ cell transcription factors, *Prdm14*, *Blimp1,* and *Tfap2c*, that are deployed by bone morphogenetic protein 4 (BMP4)-induced *TBXT*, sufficiently establish germ cell fate (*Aramaki et al., 2013*; *Nakaki et al., 2013*), whereas *SOX17* and *TFAP2C*, deployed by *EOMES* and *GATA2/3,* make up the analogous transcriptional network and fate in humans (*Kojima et al., 2017*; *Kojima et al., 2021*). Such divergence between mice and humans necessitates additional layers of caution in direct translation of IVG technologies to human infertility treatment and warrants careful scrutinization and functional validation of IVG-derived gametes in comparison to those developing naturally in vivo. Since ethical and legal constraints make research with human embryos difficult to impossible, IVG studies using model organisms that are phylogenetically close to humans is an important next step. The common marmoset (*Callithrix jacchus*) is a new-world monkey that shares many biological characteristics with humans, and thus, has been widely used for biomedical research to bridge the gap between rodent models and clinical translation (*Kishi et al., 2014*). Marmoset embryo development, including implantation and formation of fetal membranes, is well conserved with that in humans, serving as a powerful surrogate model for human post-implantation development (*Moore et al., 1985*). Moreover, the relatively short reproductive lifespan, small body size, and reasonable cost for breeding compared to other primates render the marmoset a tractable preclinical model for IVG. In particular, use of marmosets permits vigorous validation of intermediary cellular derivatives by comparing them with their in vivo counterparts (*Kishi et al., 2014*) and may enable future functional validation of resultant IVG-derived gametes by fertilization and embryo transfer.

In this study, we provide a highly efficient method to generate, expand, and maintain *C. jacchus* (cj) PGCLCs from cjiPSCs and demonstrate that these cells are immunophenotypically and transcriptionally similar to pre-migratory stage cjPGCs.

# Results

## Immunohistochemical characterization of pre-migratory cjPGCs

To validate germ cell generation in vitro, we must first have a precise understanding of the molecular features of cjPGCs in vivo. In particular, molecular characterization of early stage endogenous PGCs is critical to guide the first step of IVG – the induction of PGCLCs that appear to represent pre-migratory PGCs in humans (*Sasaki et al., 2015*; *Sasaki et al., 2016*). However, there is a dearth of information describing primate PGCs at stages before gonad colonization, primarily due to their scarcity. Therefore, we collected marmoset embryos from a triplet pregnancy at embryonic day (E)50 for immunofluorescence (IF) and molecular analyses (Carnegie stage [CS]11, 19 somites, corresponding to ~E8.5–9.0 in mice) (*Figure 1A*, *Figure 1—figure supplement 1A*). As we and others have previously identified TFAP2C, SOX17, and PDPN as specific markers of pre-migratory/migratory PGCs and PGCLCs in humans and macaque monkeys (*Sasaki et al., 2015*; *Sasaki et al., 2016*; *Sakai et al., 2020*; *Li et al., 2017*), we traced cjPGCs using these markers. At this stage, TFAP2C$^+$SOX17$^+$PDPN$^+$ cjPGCs were predominantly localized within the ventral portion of the hindgut endoderm and exhibited round nuclei with generally lower DAPI intensity (*Figure 1B and C*). A few scattered cjPGCs were also seen in the adjacent hindgut mesenchyme outside of the basement membranes, suggestive of the initiation of active migration (*Figure 1B*). Additional IF analyses revealed that cjPGCs were mostly non-proliferative (i.e., MKI67$^-$) and co-expressed pluripotency-associated markers (e.g., POU5F1, NANOG), but were negative for SOX2 (*Figure 1D and E*, *Figure 1—figure supplement 1B*, *Figure 1—source data 1A*). Notably, cjPGCs did not express later germ cell markers (e.g., DDX4 and DAZL), that are typically observed in testicular germ cells (i.e., prospermatogonia) (*Figure 1—figure supplement 1C*). IF analysis on cjPGCs showed increased global levels of histone H3 lysine 27 trimethylation (H3K27me3) and reduced global levels of histone H3 lysine 9 dimethylation (H3K9me2), consistent with the germline epigenetic reprogramming that occurs in mice, cynomolgus monkeys, and humans (*Sasaki et al., 2016*; *Tang et al., 2015*; *Seki et al., 2007*; *Figure 1F*).

## Transcriptomes of pre-migratory cjPGCs

Having identified cjPGCs residing in the hindgut endoderm by IF studies, we next set out to determine the transcriptome of endogenous cjPGCs. Given the scarcity of cjPGCs and the lack of reliable surface markers to isolate them, we first enriched cjPGCs by dissecting the posterior portions of two marmoset embryos at E50, followed by trimming of the amnion and yolk sac (*Figure 1A*). These tissues were dissociated into single-cell suspensions and subjected to high-throughput single-cell RNA-sequencing (RNA-seq) using a 10× Genomics platform. In total, 34,458 cells (6 libraries comprising 12,665 and 21,793 cells from embryos A and B, respectively) were captured for downstream analyses (*Figure 1—figure supplement 1D–I*). These cells contained a median of 2224–4198 genes/cell at a mean sequencing depth of 46–102k reads/well and 27–45% sequence saturation. To determine the cell types, we conducted hierarchical clustering and uniform manifold approximation and projection (UMAP) mapping on the combined single-cell transcriptomes from both embryos. Using known markers and differentially expressed genes (DEGs), we identified a cluster representing cjPGCs (cluster 11 marked by *TFAP2C*, *SOX17*, and *PDPN*) along with other clusters including *CLDN5*$^+$ endothelium (cluster 3), *OSR1*$^+$*PAX8*$^+$ intermediate mesoderm (cluster 4), *FOXF1*$^+$ lateral plate mesoderm (cluster 5), and *SOX2*$^+$ neutral tube (cluster 9) (*Figure 2A–C*). A full listing of all cell types that we identified and their DEGs are shown in *Figure 2B* and *Figure 2—source data 1*.

Analysis of DEGs in the cjPGC cluster revealed upregulated expression of potential germ cell specifier/regulator genes (e.g., *DND1, KIT, PRDM1, SOX15, SOX17, TFAP2C*), pluripotency-associated genes (e.g., *DPPA3, KLF4, NANOG, POU5F1, TFCP2L1, UTF1, ZFP42*), mesoderm/endoderm-associated genes (e.g., *GATA4, TBXT*), and other germ cell-related markers (*Figure 2B–E*). Accordingly, these genes were enriched with those bearing GO terms such as 'germ cell development' (*Figure 2D*). Among pluripotency-associated genes, *SOX2* was not expressed by cjPGCs, and *PRDM14* was expressed only weakly, a feature conserved with other primates (i.e., humans, cynomolgus monkeys; *Figure 2E*; *Sasaki et al., 2015*; *Sasaki et al., 2016*; *Li et al., 2017*; *Tang et al., 2015*). Expression of key proliferation markers was low, in agreement with MKI67 labeling (*Figure 1D*), suggesting that cjPGCs are largely quiescent (*Figure 1—figure supplement 1J*). Germ cell markers known to be activated upon arrival at the gonads were not expressed (e.g., *DAZL, DDX4, RNF17*) (*Figure 2E*), consistent with the pre-migratory state of these cells (*Sasaki et al., 2016*; *Li et al., 2017*).

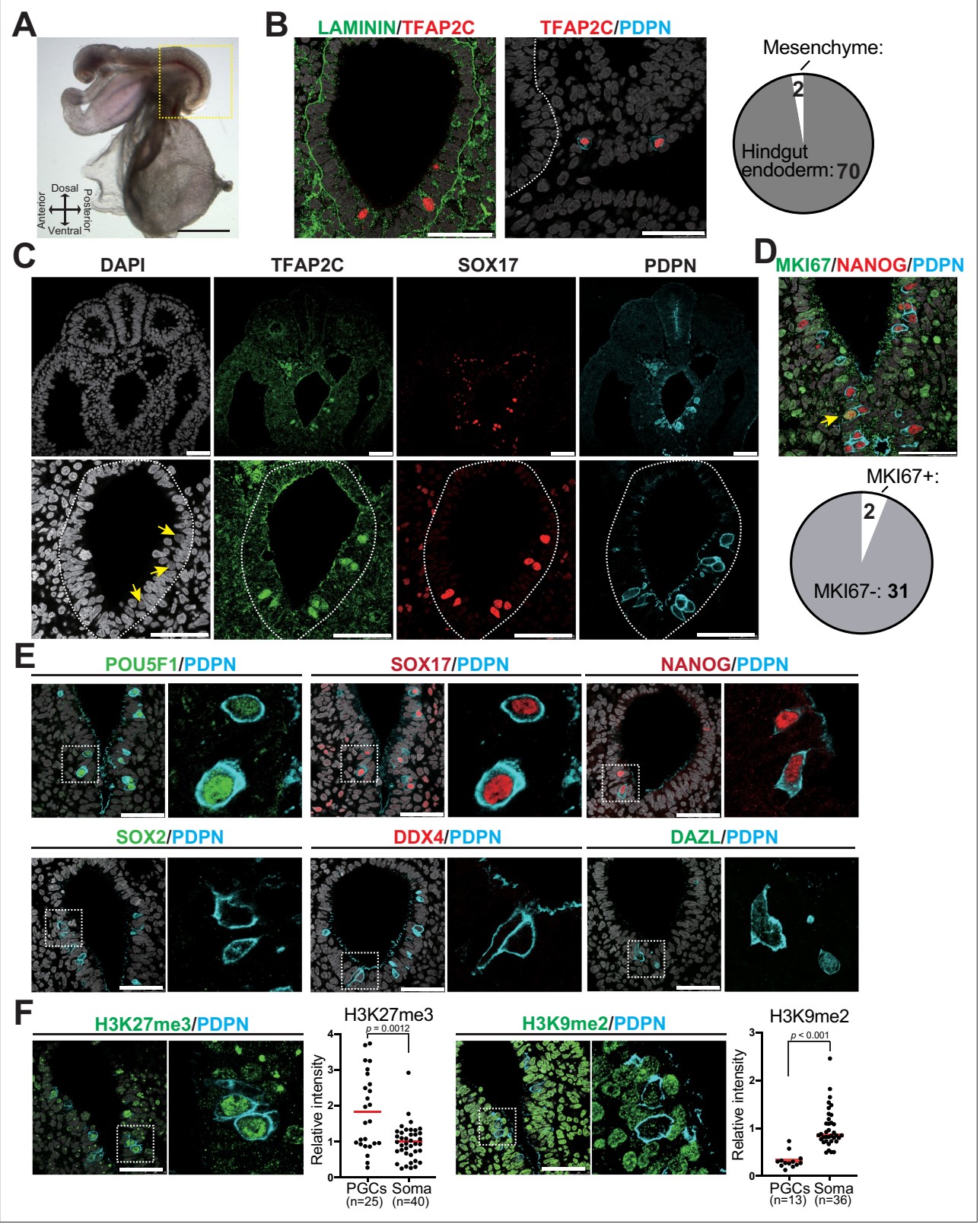

**Figure 1.** Immunophenotypic characterization of pre-migratory *Callithrix jacchus* primordial germ cells (cjPGCs) at embryonic day (E)50. (**A**) Bright field images of a cj embryo at E50 (Carnegie stage [CS]11). Scale bar, 1 mm. (**B**) (Left) Immunofluorescence (IF) images of the hindgut in the cj embryo as in (**A**) (transverse section), stained as indicated. Laminin outlines the basement membranes of the hindgut endoderm. The white dashed line highlights the hindgut endoderm. Scale bars, 50 μm. (Right) Pie chart showing the number and location of cjPGCs present in representative cross sections. (**C**) IF

*Figure 1 continued on next page*

*Figure 1 continued*

of the same cj embryo for TFAP2C (green), SOX17 (red), PDPN (cyan), and DAPI (white). Magnified images of hindgut endoderm are shown at the bottom. Arrows denote nuclei of cjPGCs with lower DAPI intensity than that of surrounding endodermal cells. Scale bar, 50 µm. (**D**) (Top) IF of the cj embryo stained for MKI67 (green), NANOG (red), and PDPN (cyan), merged with DAPI (white). An arrowhead indicates MKI67[+] cjPGC. (Bottom) Pie chart showing the number of MKI67[+] cells in PGCs. Scale bars, 50 µm. (**E**) IF of the cj embryo for pre-migratory PGC markers (POU5F1 [green], SOX17 [red], and NANOG [red]) or gonadal stage PGC markers (DDX4 [red] and DAZL [green]), co-stained for PDPN (cyan). Merged images with DAPI (white) are shown on the right of each panel. Scale bars, 50 µm. (**F**) IF of the cj embryo for PDPN (cyan), co-stained for H3K27me3 or H3K9me2 (green). Scale bars, 50 µm. Relative fluorescence intensities of H3K27me3 and H3K9me2 in PDPN[+] cjPGCs in comparison to those of surrounding somatic cells are shown on the left of each IF panel. Bar, mean. Statistical significance is determined by two-tailed Welch's t test.

The online version of this article includes the following source data and figure supplement(s) for figure 1:

**Source data 1.** Negative control images for immunofluorescence studies.

**Figure supplement 1.** Immunophenotypic and transcriptomic characterization of marmoset embryos.

**Figure supplement 1—source data 1.** Differetially expressed genes (DEGs) among cell clusters in *Figure 2B*.

In agreement with globally low H3K9me2 levels (*Figure 1F*), cjPGCs expressed low levels of enzymes for the deposition of H3K9me2 (e.g., *EHMT2*, *SUV39H1*, *SUV39H2*), and instead, expressed several H3K9 demethylases (e.g., *KDM1A/3A/3B/4A*) (*Figure 2E*). Among enzymes involved in the deposition of H3K27me3, cjPGCs expressed *EED*, *EZH2*, *SUZ12*, whereas *EZH1* expression was low (*Figure 2E*). These findings are consistent with PGCLCs/PGCs in humans and cynomolgus monkeys (*Sasaki et al., 2015*; *Sasaki et al., 2016*; *Tang et al., 2015*). Among genes related to DNA demethylation, *TET1* was expressed at high levels, whereas *TET2* and *TET3* were expressed at low levels (*Figure 2E*). *DNMT1* and *DNMT3A* were expressed at modest levels, whereas *UHRF1*, *DNMT3L*, *DNMT3B* were markedly lower, suggesting that passive demethylation might be operative due to diminished UHRF1 activity required for maintenance DNA methylation, as suggested in other species (*Sasaki et al., 2016*; *Tang et al., 2015*; *Kagiwada et al., 2013*; *Guo et al., 2015*; *Gkountela et al., 2015*).

## Derivation of cjiPSCs through peripheral blood monoculear cell reprogramming

Our next goal was to derive cjiPSCs, from which germ cells could potentially be induced. Three cell lines, 20201_6, 20201_7, and 20201_10, were established by reprogramming of peripheral blood mononuclear cells (PBMCs) (Materials and methods). Although hematological chimerism is frequently observed in marmosets (*Ross et al., 2007*; *Benirschke et al., 1962*), whole-exome sequencing confirmed that the established cjiPSCs originated from the intended PBMC donor (ID number, 38189) (*Figure 3—figure supplement 1A, B*). CjiPSCs were initially established using conventional on-feeder (OF) culture conditions (see below), but were subsequently switched to feeder-free (FF) culture conditions (PluriSTEM for basal medium and iMatrix-silk for a substrate) for its ease of maintenance. Under these conditions, FF cjiPSCs could be stably maintained over multiple passages (more than 20 passages) when passaged every 4–6 days in the presence of Y27632, a ROCK inhibitor. FF cjiPSCs bore a high nuclear to cytoplasmic ratio, were tightly packed in colonies with sharp borders and exhibited flat morphology, each of which are characteristic features of primate primed-state pluripotent cells (*Figure 3—figure supplement 1C*). These cells were mycoplasma-free, exhibited normal 46, XY karyotypes, and uniformly expressed key pluripotency-associated genes (*Figure 3—figure supplement 1D–G*).

Notably, similar to FF culture, conventional OF cultures allowed long-term propagation of cjiPSCs. However, OF cjiPSCs tended to differentiate at the center or periphery of colonies 4–5 days after passaging (*Figure 3—figure supplement 2A, B*). Moreover, OF cjiPSCs required clump passaging because single-cell passaging decreased colony formation after two passages (*Figure 3—figure supplement 2C*). Accordingly, OF cjiPSCs exhibited modest upregulation of mesodermal (e.g., *T*, *EOMES*, *MIXL1*) and endodermal genes (e.g., *FOXA2*, *SOX17*) compared to those maintained under FF conditions (*Figure 3—figure supplement 2D*). Previous studies showed that inhibition of WNT signaling stabilizes primate iPSC/ESC cultures (*Sakai et al., 2020*; *Kim et al., 2013*; *Wu et al., 2015*). Consequently, we compared our conventional OF culture to cultures containing a WNT signaling inhibitor (IWR1). Notably, OF cjiPSCs cultured with PluriSTEM containing IWR1 (OF/IWR1) maintained an undifferentiated morphology and pluripotency-associated gene expression (*Figure 3—figure*

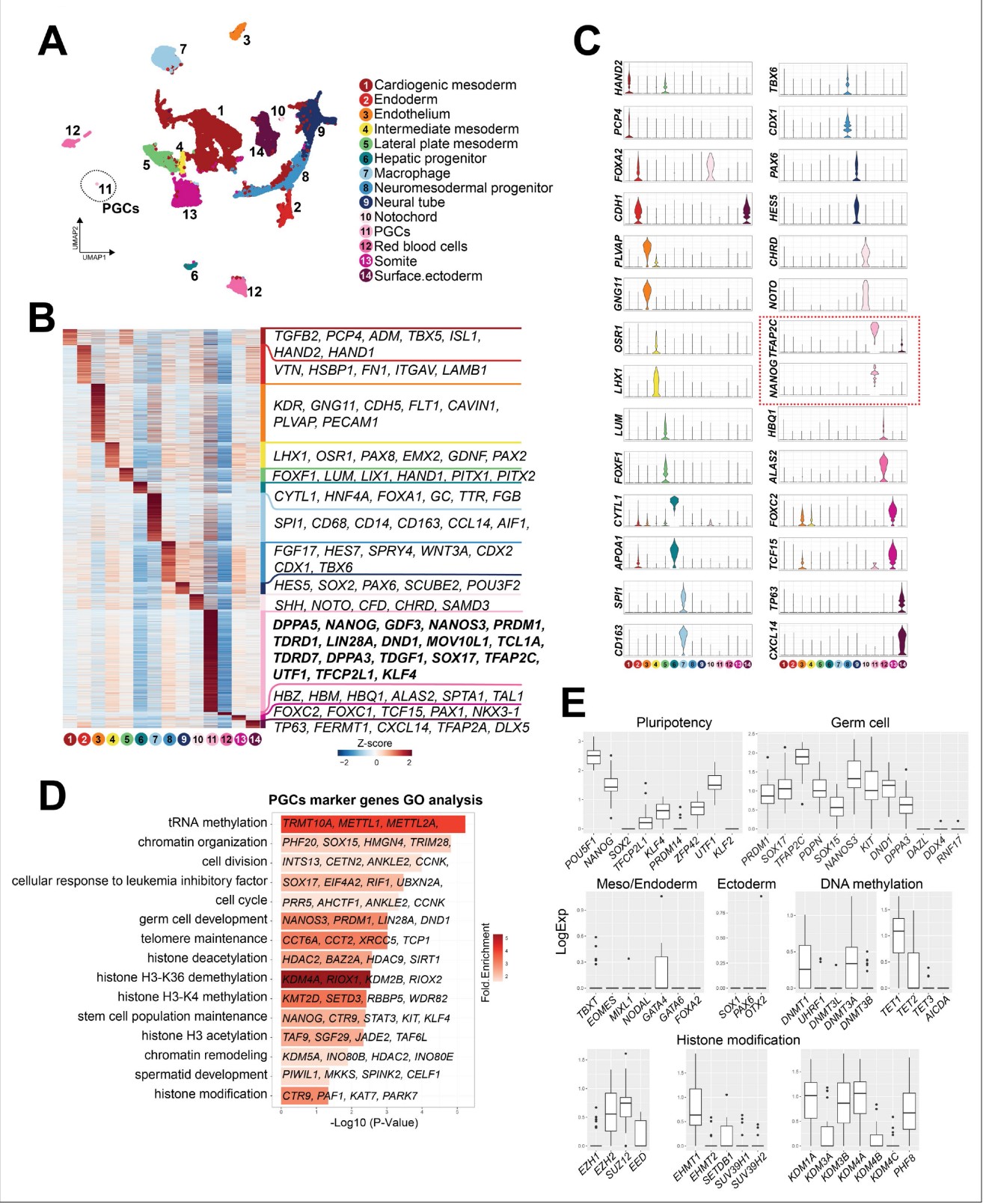

**Figure 2.** Single-cell transcriptome analyses of *Callithrix jacchus* primordial germ cells (cjPGCs) at embryonic day (E)50 (Carnegie stage [CS]11). (**A**) Uniform manifold approximation and projection (UMAP), showing different cell types in cj embryos at E50. Cell clusters are annotated on the basis of marker genes. A cluster representing cjPGCs is encircled. (**B**) Heatmap showing differentially expressed genes identified among cell types. DEGs are defined as log₂-fold change >0.25, p-value <0.01, and adjusted p-value <0.01. Representative top ranked genes are shown. (**C**) Key marker genes

*Figure 2 continued on next page*

*Figure 2 continued*

used for cell type annotation, shown as violin plots with log normalized expression. Violin plots for PGC marker genes are outlined by red dotted lines. (**D**) Gene ontology enrichment analysis of genes with significantly higher expression in cjPGCs. Bar color denotes enrichment fold changes over background. (**E**) Boxplot showing expression of key pluripotency-associated genes; germ cell, mesoderm/endoderm, and ectoderm marker genes; and DNA methylation and histone modification-associated genes. Center line, median; box limits, upper and lower quartiles; whiskers, 1.5× interquartile range.

The online version of this article includes the following source data for figure 2:

**Source data 1.** Differentially expressed genes (DEGs) among cell clusters in *Figure 2B*.

*supplement 2E, F*, *Figure 1—source data 1A*). Under this condition, mesoderm/endoderm genes were suppressed compared with conventional OF culture. Moreover, OF/IWR1 culture conditions allowed efficient single-cell passaging (*Figure 3—figure supplement 2C*). While we also found that addition of IWR1 to OF culture conditions previously utilized to grow cynomolgus monkey ESCs (AITS+IF20: advancedRPMI1640 and Neurobasal [1:1] supplemented with AllbuMax [1.6%], 1× ITS [Insulin, Transferrin, Selenium], IWR1 [2.5 μM], and bFGF [20 ng/ml]) suppressed spontaneous differentiation, the effects were not as great as when PluriSTEM was used as a basal medium (*Figure 3—figure supplement 2B, F*). Consistent with the role of Wnt inhibition in suppressing spontaneous cjiPSCs differentiation, two independent Wnt antagonists, IWR and XAV939, increased expression of pluripotency-associated surface markers, SSEA3 and SSEA4 (*Figure 3—figure supplement 2G, H*). Although cjiPSCs cultured under various conditions differed in the expression of some genes, all cjiPSCs expressed key pluripotency-associated markers (*Figure 3—figure supplement 1F*, *Figure 3—figure supplement 2D–H*), and a trilineage differentiation assay confirmed their potential to differentiate into all three germ layers (*Figure 3—figure supplement 2I, J*). Together, our findings reveal that we have identified an optimal culture protocol in both FF and OF conditions that allows stable propagation of cjiPSCs in an undifferentiated state and with a normal karyotype, thus serving as a foundation for directed differentiation toward the germline.

## Generation of PGCLCs from cjiPSCs

Our next goal was to derive cjPGCLCs directly from cjiPSCs following the protocol established in humans and cynomolgus monkeys (*Sasaki et al., 2015*; *Sakai et al., 2020*). For this, we first treated FF cjiPSCs with a PGCLC induction cocktail (i.e., BMP4, LIF, stem cell factor [SCF], epidermal growth factor [EGF], Y27632) in GK15 (Glasgow minimal essential medium [GMEM] supplemented with 15% KSR) or aRB27 (advanced RPMI1640 and supplemented with 1% B27) basal medium. Under these conditions, cjiPSCs formed aggregates with a markedly cystic appearance and did not generate SOX17⁺TFAP2C⁺ cjPGCLCs (*Figure 3—figure supplement 3A, B*), suggesting that they may not have germline competency. Thus, we next turned our attention to OF cjiPSCs without WNT inhibition given prior success in humans (*Sasaki et al., 2015*). Remarkably, upon floating culture with a PGCLC induction cocktail in GK15 or aRB27, ~3–4% PDPN⁺ITGA6^weak+ cells emerged as a distinct population starting at d4 of induction, although the frequency of such cells generally declined after d4 (*Figure 3—figure supplement 3B–D*). Sectioning of these aggregates at d4 revealed small clusters of PDPN⁺ cells uniformly expressing cjPGC markers (TFAP2C, SOX17, PRDM1, NANOG, and POU5F1), which was further confirmed by quantitative PCR (qPCR) (*Figure 3—figure supplement 3B, E, F*).

We posited that the relatively low induction efficiency of cjPGCLCs might be due to their tendency to differentiate under OF conditions. Therefore, we next utilized OF/IWR1 cjiPSCs for cjPGCLCs induction. Upon induction in floating culture, cjiPSCs readily formed tighter and more uniform size/shape aggregates compared to those induced from OF cjiPSCs (*Figure 3A and B*). Moreover, under this condition, the induction efficiency of cjPGCLCs was significantly improved, with ~15–40% cells becoming PDPN⁺ITGA6^weak+ at d4 and d6 of induction (*Figure 3C*, *Figure 3—figure supplement 3G, H*). Although variable across experiments, the median yield of PDPN⁺ cells per aggregate was ~600 at d4 and d6, but declined thereafter (*Figure 3C*). IF of sections of aggregates at d4 revealed multifocal large clusters of PDPN⁺ cells uniformly expressing key early germ cell markers (e.g., SOX17, TFAP2C, PRDM1, POU5F1, NANOG) (*Figure 3D*). Notably, this finding suggests that PDPN can serve as a highly specific surface marker of cjPGCLCs that will allow for isolation of cjPGCLCs for downstream analyses. In support, qPCR of isolated PDPN⁺ cjPGCLCs also expressed pluripotency-associated genes (i.e., *POU5F1*, *NANOG*), PGC specifier/early marker genes (i.e., *SOX17*, *TFAP2C*, *PRDM1*, and *NANOS3*)

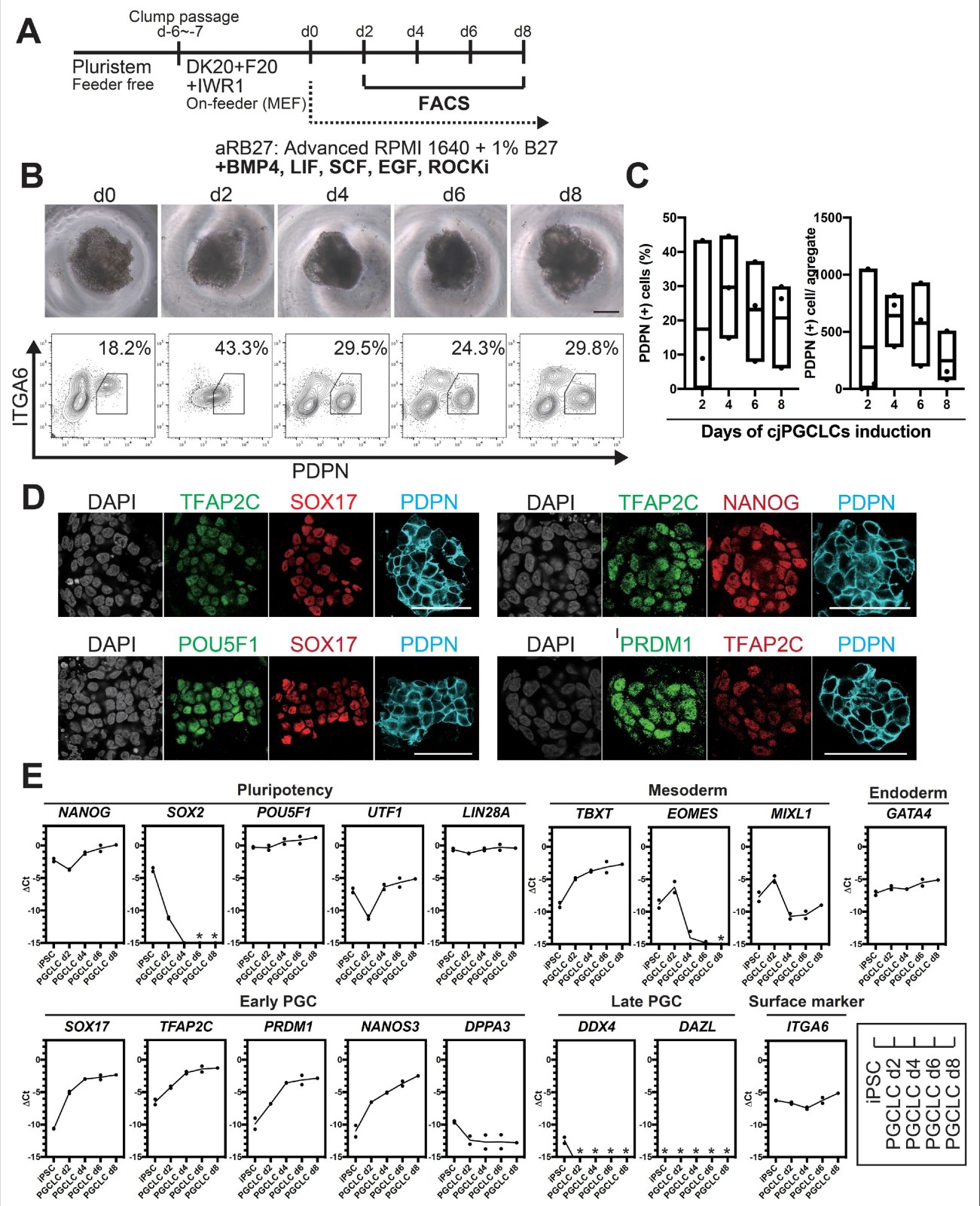

**Figure 3.** Generation of *Callithrix jacchus* primordial germ cell-like cells (cjPGCLCs) from cj induced pluripotent stem cells (cjiPSCs). (**A**) Scheme for cjPGCLC induction. (**B**) BF images (top) and fluorescence-activated cell sorting (FACS) plots (bottom) for the floating aggregates of cjiPSCs induced to differentiate into cjPGCLCs. The percentages of PDPN⁺ITGA6ʷᵉᵃᵏ⁺ cells are shown. Scale bars, 200 μm. (**C**) Boxplot representations of the induction kinetics of PDPN⁺ITGA6ʷᵉᵃᵏ⁺ cells (left, percentages; right, number of cells/aggregate) during PGCLC induction in aRB27. Center line, median; box

*Figure 3 continued on next page*

*Figure 3 continued*

limits, upper and lower quartiles; whiskers, 1.5× interquartile range. (**D**) Immunofluorescence (IF) images of floating aggregates after 6 days of PGCLC induction, stained as indicated. Scale bars, 50 µm. (**E**) Gene expression of cjiPSCs and cjPGCLCs at days 2, 4, 6, and 8, as measured by quantitative PCR (qPCR). For each gene examined, the ΔCt values were derived using the average Ct values of the two housekeeping genes *GAPDH* and *PPIA* (set as 0) calculated and plotted for two independent experiments. *Not detected.

The online version of this article includes the following figure supplement(s) for figure 3:

**Figure supplement 1.** Derivation and feeder free culture of *Callithrix jacchus* induced pluripotent stem cells (cjiPSCs).

**Figure supplement 2.** Culture of *Callithrix jacchus* induced pluripotent stem cells (cjiPSCs) on a feeder layer with an inhibitor of WNT signaling.

**Figure supplement 3.** Induction of *Callithrix jacchus* primordial germ cell-like cells (cjPGCLCs) from OF, OF/IWR1, or FF cj induced pluripotent stem cells (cjiPSCs).

and lacked detectable *SOX2* and late germ cell marker (i.e., *DDX4*, *DAZL*) (*Figure 3E*), features similar to pre-migratory PGCs (*Figures 1D and 2E*; *Sasaki et al., 2016*). Together these results indicate that our in vitro platform enables highly efficient and reproducible generation of cjPGCLCs.

## 2D cjPGCLCs expansion culture

Induction of cjPGCLCs from cjiPSCs via floating aggregates is somewhat time-consuming and limited in scalability. As such, 2D expansion of PGCLCs that retain the cellular and molecular characteristics of PGCs would greatly enhance our ability to generate PGCLCs in a scalable manner that can be utilized, off-the-shelf, for downstream molecular and functional characterization. To accomplish this, we modified a culture method previously utilized to expand human (h)PGCLCs (*Figure 4A*; *Murase et al., 2020*). Specifically, we cultured sorted d6 PDPN+ cjPGCLCs on a STO-feeder layer in DK15 medium containing 2.5% fetal bovine serum (FBS), SCF, FGF2, and forskolin. Plated cjPGCLCs formed loosely arranged clusters, which increased in size and became confluent by expansion culture day (c)10 (*Figure 4B*). These cells expressed markers of early cjPGC/PGCLCs but did not possess late germ cell markers (i.e., *DDX4*, *DAZL*), suggesting that they retain the cellular state of cjPGCLCs (*Figure 4C–E*, *Figure 1—source data 1A*). Moreover, these cells could be passaged approximately every 10 days by dissociation and fluorescence-activated cell sorting (FACS) of PDPN+ cells and exhibited exponential growth for at least 30 days (*Figure 4F and G*). Although marker expression pattern was largely unchanged during 30 days of expansion culture, *DPPA3* showed modest upregulation, similar to hPGCLCs under expansion culture (*Figure 4H*). *ITGA6*, which is a surface marker weakly expressed on cjPGCLCs, also exhibited modest upregulation along the time course (*Figure 4H*). These findings highlight the feasibility of 2D expansion culture of cjPGCLCs analogous to hPGCLCs.

## Maturation of cjPGCLCs into early prospermatogonia-like state

One of the functional features of PGCLCs is their capacity to further develop into more advanced germ cells (*Hwang et al., 2020*; *Yamashiro et al., 2018*; *Hayashi et al., 2012*; *Hayashi et al., 2011*). Therefore, we next utilized a xenogeneic reconstituted testis culture that allows hPGCLCs to mature into prospermatogonia to determine if cjPGCLCs could similarly differentiate (*Hwang et al., 2020*). After expansion of cjPGCLCs for 30 days by 2D culture, we initiated an xrTestis culture by mixing sorted PDPN+ cjPGCLCs with mouse fetal testicular somatic cells depleted of endogenous germ cells (*Figure 5A*). After 2 days of floating culture, xenogeneic reconstituted testes (xrTestes) formed tight aggregates, which were subsequently maintained by air-liquid interface (ALI) cultures (*Figure 5A and B*). At day 15 of ALI culture, we observed reconstituted testicular cords surrounded by NR2F2+ interstitial cells in xrTestis cultures (*Figure 5C*, *Figure 1—source data 1A*). Notably, there were a number of TFAP2C+POU5F1+NANOG+ cjPGCLCs, which primarily localized peripheral to SOX9+ mouse-derived Sertoli cell nuclei (*Figure 5C*). In addition, xrTestes maintained until day 30 of ALI culture revealed prominent proliferation of TFAP2C+ germ cells, which forced SOX9+ Sertoli cells toward the center of the testicular cords (*Figure 5C*). Remarkably, we found a few scattered DAZL+DDX4+SOX-17+POU5F1+TFAP2C+SOX2- cells (DDX4+TFAP2C+ cells [4/123, 3.3% among all TFAP2C+ cells] and DAZL+TFAP2C+ cells [2/232, 0.86% among all TFAP2C+ cells]), suggesting progression into early prospermatogonia (*Figure 5C*; *Hwang et al., 2020*). Together, these data indicate that cjPGCLCs can be integrated in the testicular niche and are capable of further expansion and differentiation.

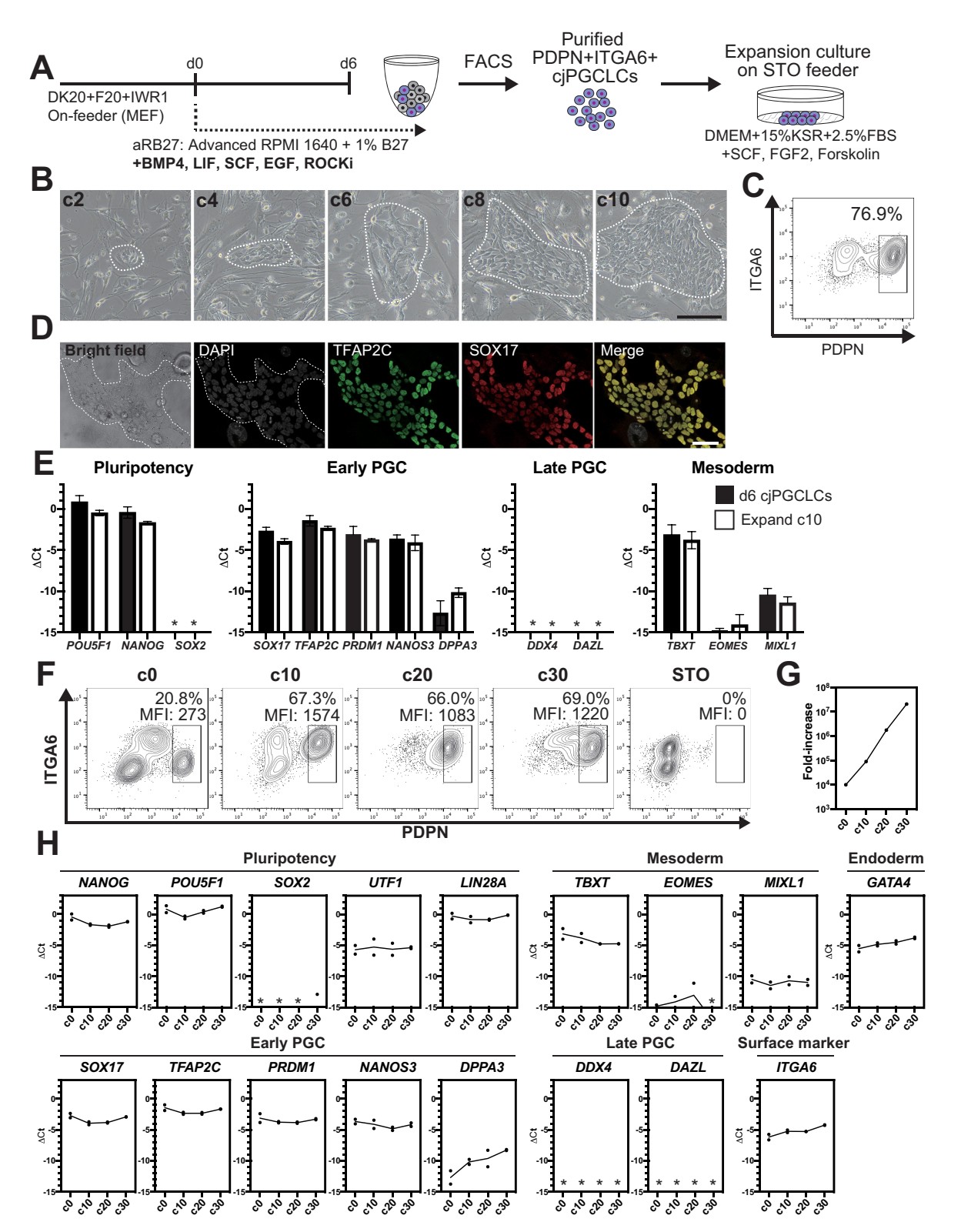

**Figure 4.** 2D expansion culture of *Callithrix jacchus* primordial germ cell-like cells (cjPGCLCs). (**A**) Scheme for expansion culture of cjPGCLCs. (**B**) BF images of c2, c4, c6, c8, and c10 colonies of cjPGCLCs. The white dashed lines highlight colonies of cjPGCLCs. Scale bars, 200 μm. (**C**) Fluorescence-activated cell sorting (FACS) analysis of c10 expansion cultures of cjPGCLCs. The percentage of PDPN⁺ITGA6 ⁺ cells is shown. (**D**) Immunofluorescence (IF) images of expansion culture day 10 (c10) cjPGCLCs for DAPI (white), TFAP2C (green) and SOX17 (red), and the merged image. Scale bars, 50 μm.

*Figure 4 continued on next page*

*Figure 4 continued*

(**E**) Gene expression of d6 cjPGCLCs and c10 cjPGCLCs, as measured by quantitative PCR (qPCR). For each gene examined, the ΔCt values from the average Ct values of the two housekeeping genes *GAPDH* and *PPIA* (set as 0) were calculated and plotted for two independent experiments. *, not detected. (**F**) FACS analyses of c0 (d6 cjPGCLCs), c10, c20, c30, and c40 cjPGCLCs. The percentages and mean fluorescence intensity (MFI) of PDPN⁺ITGA6 ⁺ cells are shown. (**G**) Growth curve of PDPN⁺ITGA6⁺ cells during cjPGCLC expansion culture until c30. A total of 10,000 PDPN⁺ITGA6$^{weak+}$ d6 PGCLCs were used as a starting cell population. (**H**) qPCR analyses of the expression of the indicated genes during cjPGCLC expansion culture. Mean values are connected by a line. *, not detected.

## Transcriptome accompanying formation of cjPGCLCs

We next sought to define gene expression dynamics accompanying specification of cjPGCLCs by bulk RNA-seq (*Figure 6—figure supplement 1A*). Unsupervised hierarchical clustering (UHC) classified the cells during cjPGCLCs induction largely into two clusters, one with FF, FF/IWR1, and OF cjiPSCs and the other with cjPGCLCs and OF/IWR1 cjiPSCs, which was also supported by Pearson correlation among clusters (*Figure 6A and B*). The relative positioning of cjPGCLC samples in principal component (PC) space supports a stepwise developmental progression during the in vitro culture (*Figure 6C*). First, FF and FF/IWR1 cjiPSCs were intermingled and formed a discrete cluster that was most distinct from cjPGCLCs. There were no significant differences in gene expression between FF and FF/IWR1 cjiPSCs, suggesting that IWR1 does not significantly alter the cellular properties of FF cjiPSCs (*Figure 6C*, *Figure 6—figure supplement 1B*). Notably, OF and OF/IWR1 cjiPSCs were positioned closer to cjPGCLCs in PC space, with OF/IWR1 cjiPSCs being closest to d2 cjPGCLCs, consistent with their higher competency to differentiate into cjPGCLCs (*Figure 6C*). OF/IWR1 cjiPSCs bore gene expression signatures characteristic of primed-state pluripotency, similar to that seen in FF or OF cjiPSCs (*Figure 6—figure supplement 1C*; *Nakamura et al., 2016*). Notably, while most key germ cell genes were not significantly upregulated, there is a modest upregulation of *TFAP2C* and *PRDM14* in OF/IWR1 cjiPSCs, which might contribute to their high germline competency (*Figure 6—figure supplement 1C*).

Pairwise comparison of gene expression revealed that genes were primarily upregulated as FF cjiPSCs transitioned to OF and OF/IWR1 cjiPSCs (*Figure 6—figure supplement 1D*). GO terms among the enriched genes in OF and OF/IWR1 cjiPSCs included 'protein destabilization' or 'ubiquitin-dependent protein catabolic process' (*Figure 6—figure supplement 1D*). Expression of most of these genes was sustained until d2 cjPGCLCs, suggesting that changes associated with ubiquitin-proteasome system (UPS)-mediated protein turnover might confer a permissive cellular environment for cjPGCLCs specification (*Figure 6—figure supplement 1E, F*). Clustering analysis of variably expressed genes across the developmental trajectory revealed four large clusters (*Figure 6D*). Genes in cluster 1 represented those with relatively high expression in cjiPSCs, but which are downregulated during differentiation into cjPGCLCs. Those genes were enriched with GO terms such as 'inner cell mass cell proliferation' or 'stem cell population maintenance', consistent with their pluripotent nature (*Figure 6D*). Genes in cluster 2 were those upregulated along the trajectory and included key germ cell genes (e.g., *DND1*, *NANOS3*, *PRDM1*, *SOX17*, *TFAP2C*) and GO terms included 'germ cell development'. Moreover, GO terms such as 'DNA methylation' or 'histone methylation' were also seen, consistent with the dynamic epigenetic remodeling observed in developing PGCs. Genes in cluster 3 were those primary upregulated in 2D expansion culture cjPGCLCs and included an enriched GO term, 'response to oxidative stress', which might suggest changes associated with culture adaptation. Finally, genes in cluster 4 were those transiently upregulated in d2 cjPGCLCs. These genes included endoderm and mesoderm markers (e.g., *EOMES*, *HAND1*, *MESP1*, *MIXL1*, *NODAL*, *SNAL1*) and were enriched with GO terms such as 'mesoderm formation' and 'cellular response to BMP stimulus', suggesting that cjPGCLC induction may be transiently accompanied by somatic programs, as previously observed following PGCLC induction in other primates (*Figure 6D–F*; *Sasaki et al., 2015*; *Sakai et al., 2020*; *Irie et al., 2015*; *Kobayashi et al., 2017*).

We next evaluated the dynamics of gene expression associated with germ cell specification and development. We noted that key germ cell specifier genes (e.g., *SOX17*, *TFAP2C*, *PRDM1*, *NANOS3*) started to increase in expression and *SOX2* was swiftly downregulated in d2 cjPGCLCs (*Figure 6G*, *Figure 6—figure supplement 1D*, *Figure 6—figure supplement 2*). *TBXT*, which is only transiently activated in mPGCLCs, continued to be expressed after d2, similar to hPGCLCs (*Figure 6G*; *Sasaki et al., 2015*; *Sakai et al., 2020*). Notably, *DDX4* and *DAZL*, germ cell markers expressed upon entry

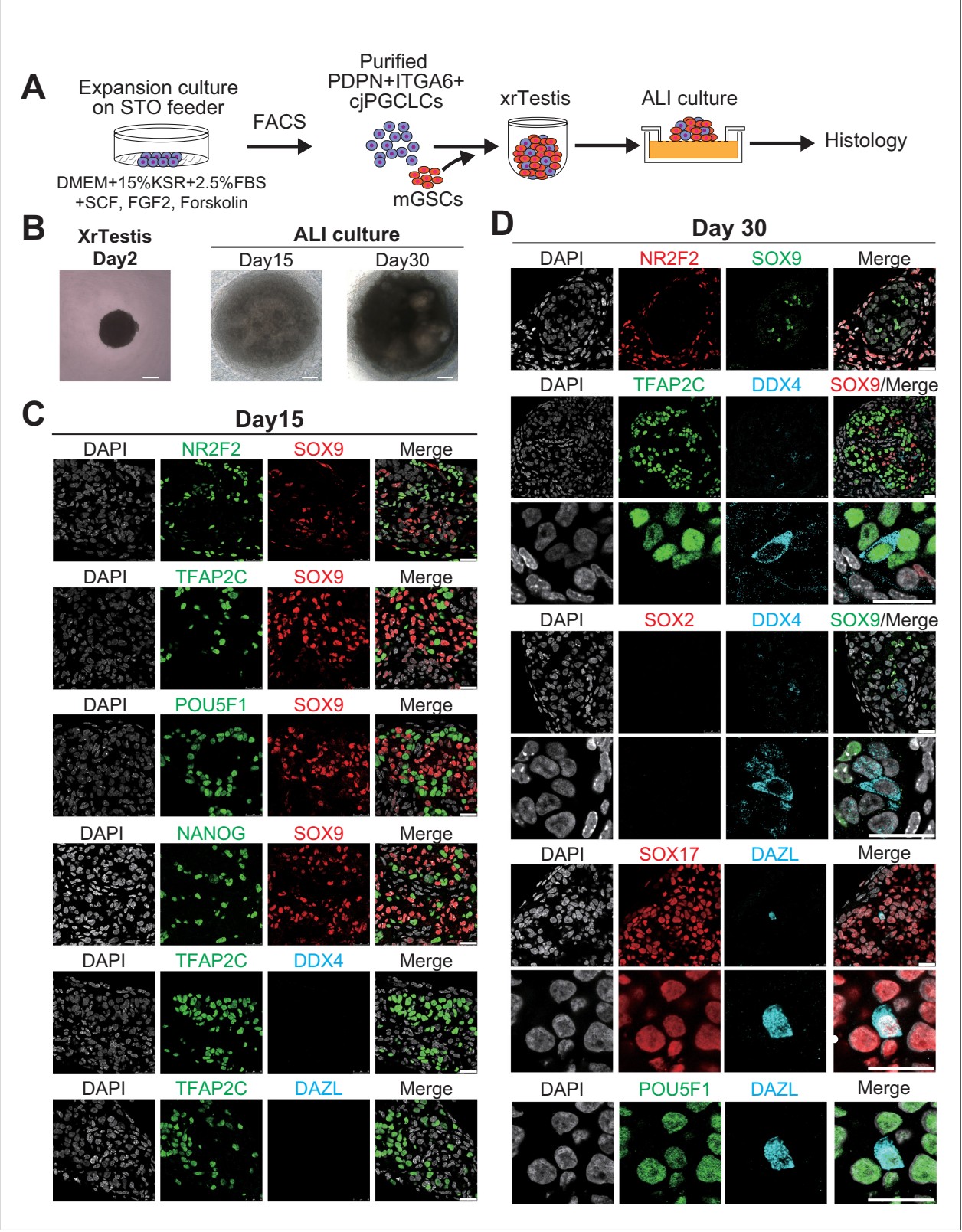

**Figure 5.** Maturation of *Callithrix jacchus* primordial germ cell-like cells (cjPGCLCs) into a DDX4⁺ prospermatogonia-like state. (**A**) Scheme for xrTestis culture. ALI, air-liquid interphase; xrTestis, xenogeneic reconstituted testis; mGSOs, mouse gonadal somatic cells derived from E12.5 mouse embryonic testes depleted of endogenous germ cells. (**B**) Bright field images of d15 and d30 xenogeneic reconstituted testes (xrTestes) ALI culture. Bar, 200 μm. (**C**) (Left) Immunofluorescence (IF) images of d15 (left) xrTestes, showing expression of the indicated key PGC markers (TFAP2C, POU5F1, and NANOG

*Figure 5 continued on next page*

*Figure 5 continued*

[green]), somatic cell markers (SOX9, Sertoli cell marker [red]; NR2F2, interstitial cell marker [green]), or a gonadal stage germ cell marker, DDX4 (cyan). (**D**) IF images of d30 xrTestes, indicating expression of the key primate PGC markers (TFAP2C, POU5F1, SOX17), a mouse PGC marker (SOX2), prospermatogonial markers (DDX4, DAZL) or somatic cell markers (NR2F2, SOX9). Merged images with DAPI are shown on the right. Scale bars, 50 μm.

into the gonad, were not expressed (*Figure 6G*), consistent with their lack of expression in pre-migratory cjPGCs at E50 (*Figures 1E and 2E*).

## scRNA-seq revealed lineage trajectory and gene expression dynamics during formation of cjPGCLCs

To better define the lineage trajectory and transcriptomic dynamics accompanying cjPGCLC formation, we performed scRNA-seq on three samples (OF/IWR1 cjiPSCs, d2 and d6 cjPGCLCs [whole aggregates without FACS]). After QC validation and filtering, 9098 cells were used for downstream analysis (*Figure 7—figure supplement 1A*). Transcriptomes of cells were aggregated and projected onto a t-distributed stochastic neighbor embedding (tSNE) after dimension reduction and clustering, which yielded six clusters (*Figure 7—figure supplement 1B–D*). These clusters were annotated based on marker gene expression and DEGs (*Figure 7—figure supplement 1B–F*). As expected, we observed various off-target cells (cardiac, endoderm or endothelial lineages, macrophages) and a small fraction of apoptotic cells (*Figure 7—figure supplement 1F*). Notably, we also identified a cell cluster that expressed pluripotency-associated (e.g., *NANOG*, *POU5F1*) and/or germ cell specifier genes (e.g., *SOX17*, *TFAP2C*), with its DEGs enriched with GO term such as 'germ cell development' or 'stem cell maintenance' (*Figure 7—figure supplement 1E, F*). This cluster primarily contains cjiPSCs and germline lineage and further subsetting revealed six subclusters (clusters 3, 1/4/6, and 2/5, consisting primarily of cjiPSCs, d2 or d6 cjPGCLCs, respectively) (*Figure 7A and B*, *Figure 7—figure supplement 1G, H*). When projected on PHATE embedding, these subclusters aligned along the actual sample stages and pseudotime trajectory (*Figure 7A–D*), a finding further supported by RNA velocity analysis (*Figure 7E*). Accordingly, genes related to stem cell maintenance were downregulated along this trajectory, suggestive of differentiation from cjiPSCs (GO terms include 'stem cell population maintenance'). In contrast, early germ cell marker/specifier genes (e.g., *KIT*, *NANOS3*, *DND1*, *SOX17*, *TFAP2C*) or genes related to chromatin remodeling were upregulated, suggestive of germ cell specification and accompanying epigenetic remodeling (*Figure 7F–H*). Consistent with bulk RNA-seq analyses, a number of mesodermal genes (e.g., *EOMES*, *HAND1*, *MIXL1*) were transiently upregulated in clusters 1/4/6, which were enriched with GO terms such as 'mesoderm formation' (*Figure 7F and G*, *Figure 7—figure supplement 1H*). In contrast, many pluripotency-associated genes (e.g., *NANOG*, *UTF1*) were transiently downregulated during this transition, similar to mouse but not human PGCLC specification (*Figure 7F and H*; *Sasaki et al., 2015*; *Hayashi et al., 2011*).

We next assessed the transcriptional similarities and differences between cjPGCLCs in vitro and E50 cjPGCs in vivo. As expected, E50 cjPGCs (cluster 7) juxtaposed with clusters 2/5 (d6 cjPGCLCs) in PHATE embedding (*Figure 7I–K*), and these clusters revealed a high correlation by Pearson correlation analysis (*Figure 7L*). Pairwise correlation analysis revealed that many key germ cell markers/specifier genes are equally upregulated in both cell types (*Figure 7M*). Notably, we found that d6 cjPGCLCs were more proliferative/apoptotic whereas E50 cjPGCs upregulated genes were enriched in 'regulation of transcription from RNA polymerase II promoter' (*Figure 7M*). Some up- and down-regulated DEGs were enriched with GO terms 'actin cytoskeleton organization' (up in d6 cjPGCLCs) or 'response to insulin' (up in E50 cjPGCLCs), which might reflect culture adaptation or differences in the surrounding nutritional/hormonal environment (*Figure 7M*).

Finally, we made a cross-species comparison of PGCLC genes (genes upregulated in PGCLCs compared to iPSCs) between marmoset and humans. We found that a large fraction of PGCLC genes were conserved, including genes associated with transcriptional regulation (most key germ cell specifier genes [e.g., *TFAP2C*, *SOX17*] were included in this category) or protein ubiquitination (*Figure 7—figure supplement 2A–C*). Notably, species-specific PGCLC genes were enriched in GO terms such as 'fatty acid homeostasis' (human PGCLC upregulated) or 'carbohydrate metabolic process' (marmoset PGCLC upregulated), which might indicate a species-specific metabolic requirement in early developing germ cells (*Figure 7—figure supplement 2C*).

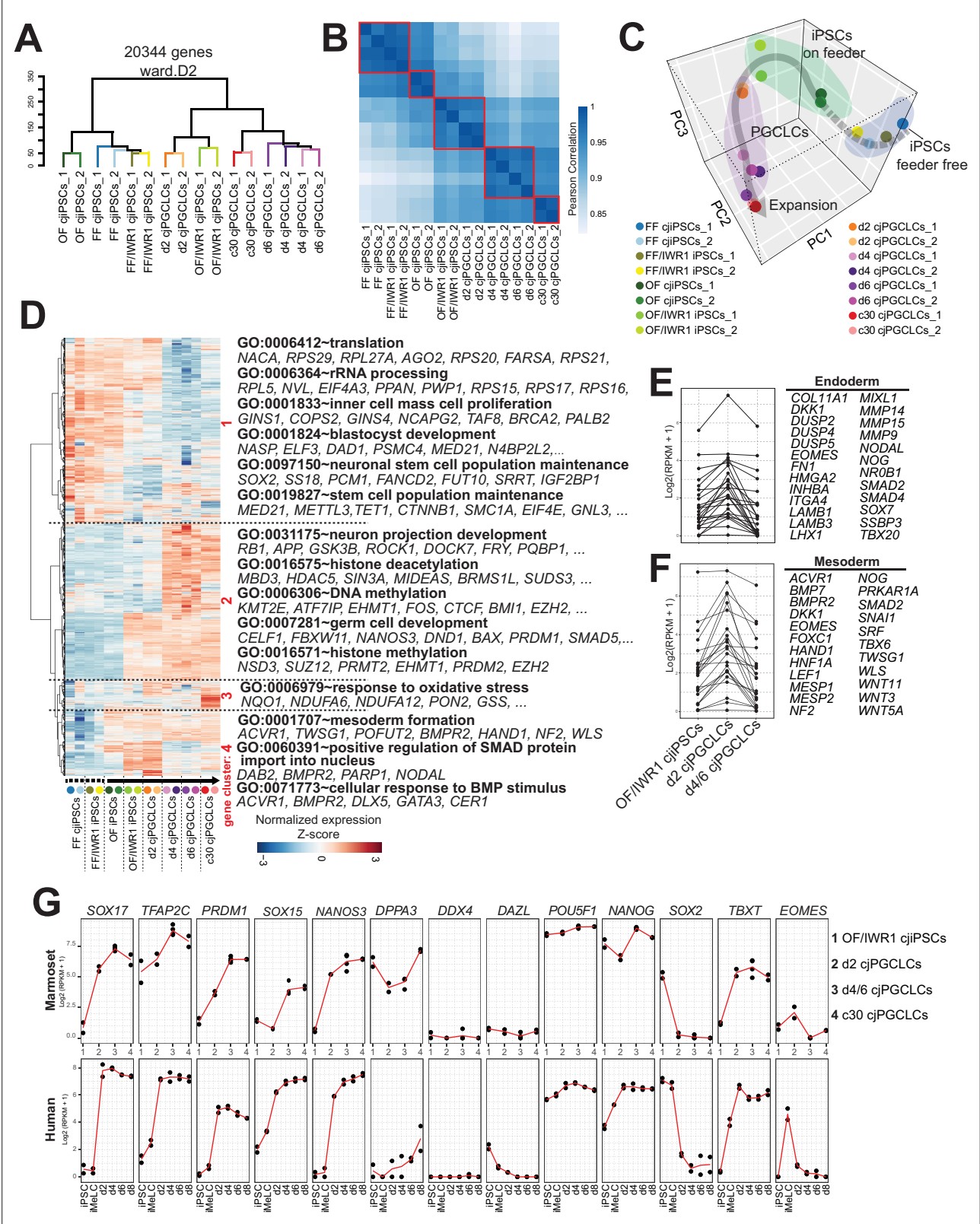

**Figure 6.** Transcriptome accompanying formation of *Callithrix jacchus* primordial germ cell-like cells (cjPGCLCs). (**A**) Unsupervised hierarchical clustering (UHC) of the transcriptomes of all samples by using ward.D2. (**B**) Pearson correlation of samples as in (**A**). Highly correlated samples are encircled with red lines. (**C**) Principal component analysis (PCA) of the samples used in this study. The gray arrow represents a trajectory for cjPGCLC specification. (**D**) UHC of the top 5000 variably expressed genes among samples, which are largely divided into four gene clusters (highlighted in red,

*Figure 6 continued on next page*

*Figure 6 continued*

roughly categorized as: 1, genes higher in cjiPSCs; 2, genes higher in cjPGCLCs; 3, genes higher in expansion culture cjPGCLCs; 4, genes higher in d2 cjPGCLCs). The gene expression level is represented by a heatmap. Samples were aligned along the estimated trajectory as defined in (**C**). Gene expression is row scaled with colors indicating the Z-score. Enriched gene ontology (GO) terms and representative genes in each gene cluster are labeled beside the heatmap. (**E, F**) Expression of endoderm (**E**) or mesoderm (**F**) genes during the transition of OF/IWR1 cjiPSCs to d4/6 cjPGCLCs. These genes were selected according to GO terms (GO:0001706, endoderm formation; GO:0001707, mesoderm formation). (**G**) Gene expression dynamics during cjPGCLC induction and c30 expansion culture, as measured by quantitative PCR (qPCR) (top). For comparison, gene expression dynamics during human PGCLC induction is also shown (bottom). During induction of cjPGCLCs in vitro, key genes showed expression patterns similar to those seen during human PGCLC induction (*Sasaki et al., 2015*). Expression is normalized by $\log_2$(RPKM + 1).

The online version of this article includes the following source data and figure supplement(s) for figure 6:

**Source data 1.** Top 5000 variably expressed genes among transcriptomes of different types of cells in *Figure 6D*.

**Figure supplement 1.** Transcriptomic dynamics associated with *Callithrix jacchus* primordial germ cell-like cell (cjPGCLC) induction.

**Figure supplement 1—source data 1.** Differentially expressed genes (DEGs) from pairwise comparisons in *Figure 6—figure supplement 1D*.

**Figure supplement 2.** Dynamics of key marker gene expression associated with *Callithrix jacchus* primordial germ cell-like cell (cjPGCLC) induction.

## Global DNA methylation in cjPGCLCs

Previous studies suggested that hPGCLCs showed only modest reductions in global 5mC levels with or without expansion culture, suggesting that hPGCLCs have just undergone specification and have not yet completed global DNA demethylation, a hallmark of mammalian PGC development (*Saitou and Miyauchi, 2016*). Therefore, we next evaluated global 5mC levels in cjPGCLCs by whole-genome bisulfite sequencing (WGBS). Similar to hPGCLCs (*Sasaki et al., 2015*; *Murase et al., 2020*), d4 PGCLCs showed a slight but significant reduction in 5mC levels (mean, ~63%) compared with OF/IWR1 cjiPSCs (mean, ~75%) (*Figure 8A and B*). Notably, cjPGCLCs in expansion culture exhibited further reduction in 5mC levels, bearing a 5mC level of ~50% at c30 (*Figure 8A and C*). Thus, the dynamics of global 5mC levels during cjPGCLC induction and expansion is similar to that of humans (*Sasaki et al., 2015*; *Murase et al., 2020*). To gain further insight into the regulation of global DNA methylation profiles in cjPGCLCs, we evaluated expression dynamics of genes related to DNA methylation. Among de novo DNA methyl transferases, *DNMT3B* was highly expressed in cjiPSCs, but exhibited a sharp downregulation upon cjPGCLC induction (*Figure 8D*). On the other hand, *DNMT3A* showed modest downregulation upon cjPGCLC induction, and *DNMT3L* was expressed only at low levels in all cells examined. Among the genes related to maintenance of DNA methylation, *DNMT1* was expressed at a significant level in all cells analyzed, whereas *UHRF1*, which is responsible for the recruitment of *DNMT1* into replication foci (*Kagiwada et al., 2013*; *Sharif et al., 2007*; *Bostick et al., 2007*), showed a marked reduction upon cjPGCLC induction (*Figure 8D*). In cjPGCLC expansion cultures, *DNMT3B* expression was further downregulated whereas *UHRF1* showed slightly higher expression than d4/6 cjPGCLCs (*Figure 8D*). Among genes related to active DNA demethylation, *TET1* was highly expressed in all cells whereas expression of *TET2* and *TET3* were very low. Thus, compared to cjiPSCs, cjPGCLCs at d6 or in expansion culture showed reduced but detectable levels of *DNMT3B* and *UHRF1*, which might serve as a basis for the modest reduction of global DNA methylation of these cells. In mice, EHMT2 (~E7.5 onward) followed by EHMT1 (~E9.5 onward) are downregulated in PGCs, which might account for low global H3K9Me2 levels. Similar to human PGCLCs, we found that cjPGC/cjPGCLCs both show marked downregulation, which might explain the low global H3K9me2 levels. Overall, the expression pattern of epigenetic modifiers, including those related to DNA methylation, is similar to those observed in endogenous cjPGCs at E50 (*Figure 2E*), further supporting the notion that cjPGCLCs derived by our protocol accurately resemble endogenous pre-migratory cjPGCs.

## Discussion

In contrast to the relatively well-characterized germ cell development of *C. jacchus* postnatally, there is a paucity of information regarding the transcriptomic and epigenomic properties at earlier stages, primarily due to the inherent difficulty in recovering marmoset embryos. Previous studies demonstrated the presence of POU5F1[+]NANOG[+] cjPGCs localized within the hindgut endoderm in an E50 embryo, similar to that observed in human and monkey embryos at equivalent stages (*Aeckerle*

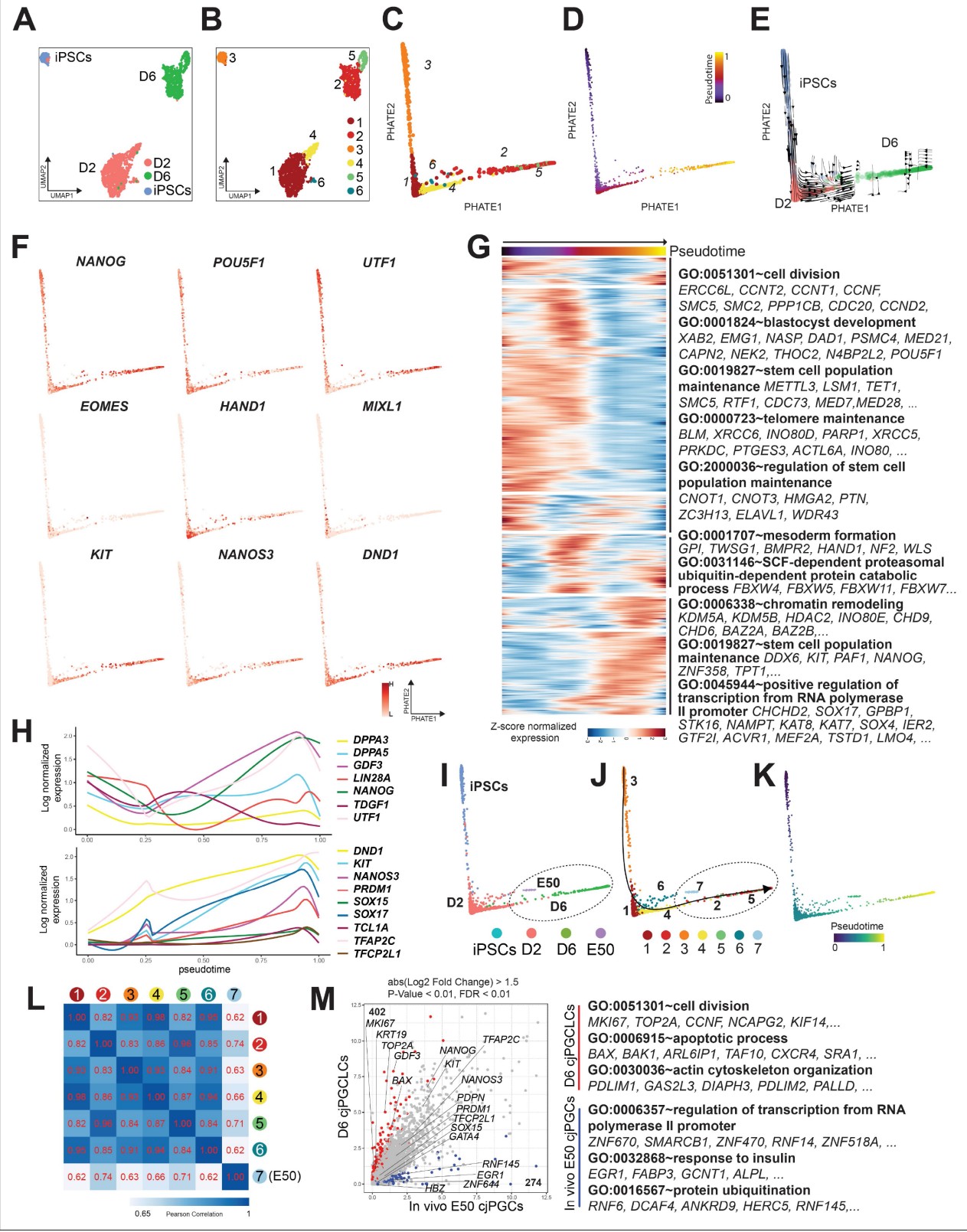

**Figure 7.** scRNA-seq revealed lineage trajectory and gene expression dynamics during formation of *Callithrix jacchus* primordial germ cell-like cells (cjPGCLCs). (**A, B**) Uniform manifold approximation and projection (UMAP) plots showing all cells annotated as 'pluripotent/germ' defined in *Figure 7—figure supplement 1B*, colored according to sample origin (**A**) or subclusters (clusters 1–6) (**B**). (**C, D**) Trajectory analysis of transcriptomes as in (**B**) projected to PHATE embedding. Cells were colored according to cell clusters (**C**) or pseudotime (**D**). (**E**) PHATE embedding of transcriptomes as

*Figure 7 continued on next page*

*Figure 7 continued*

in (**C**) with overlaid RNA velocity. Cells were colored according to sample origin. (**F**) Expression of key pluripotency-associated genes (top), mesodermal genes (middle), or germ cell markers (bottom) projected on PHATE embedding as in (**C**). (**G**) Transcriptome dynamics along the pseudotime trajectory as in (**D**). The top 5000 highly variable genes are hierarchically clustered with three different patterns along pseudotime. Enriched gene ontology (GO) terms are listed at right. Each row is a gene, and each column is a cell ordered by pseudotime. Expression is log normalized and scaled by row. (**H**) Expression dynamics of key pluripotency-associated genes (top) or germ cell marker/specifier genes (bottom) aligned along pseudotime as in (**D**). (**I**–**K**) Trajectory analysis of transcriptomes as in (**B**) combined with those of E50 cjPGCs projected on PHATE embedding, colored according to sample origin (**I**), cell cluster (**J**), or pseudotime (**K**). Cluster 7 consists exclusively of E50 cjPGCs. (**L**) Pearson correlation of transcriptomes of E50 cjPGCs (cluster 7) and other in vitro-derived clusters (clusters 1–6) as in (**B**). (**M**) Scatter plot comparison of the averaged expression values of differentially expressed genes (DEGs) between E50 cjPGCs (cluster 7) and d6 cjPGCLCs (clusters 2/5). DEGs are defined as log$_2$ fold change above 1.5 (p-value <0.01 and FDR <0.01). Over-represented GO terms and representative genes in each GO category are shown on the right.

The online version of this article includes the following figure supplement(s) for figure 7:

**Figure supplement 1.** Single-cell transcriptomes during *Callithrix jacchus* primordial germ cell-like cell (cjPGCLC) induction.

**Figure supplement 2.** Comparison of primordial germ cell-like cells (PGCLCs) between humans and marmosets.

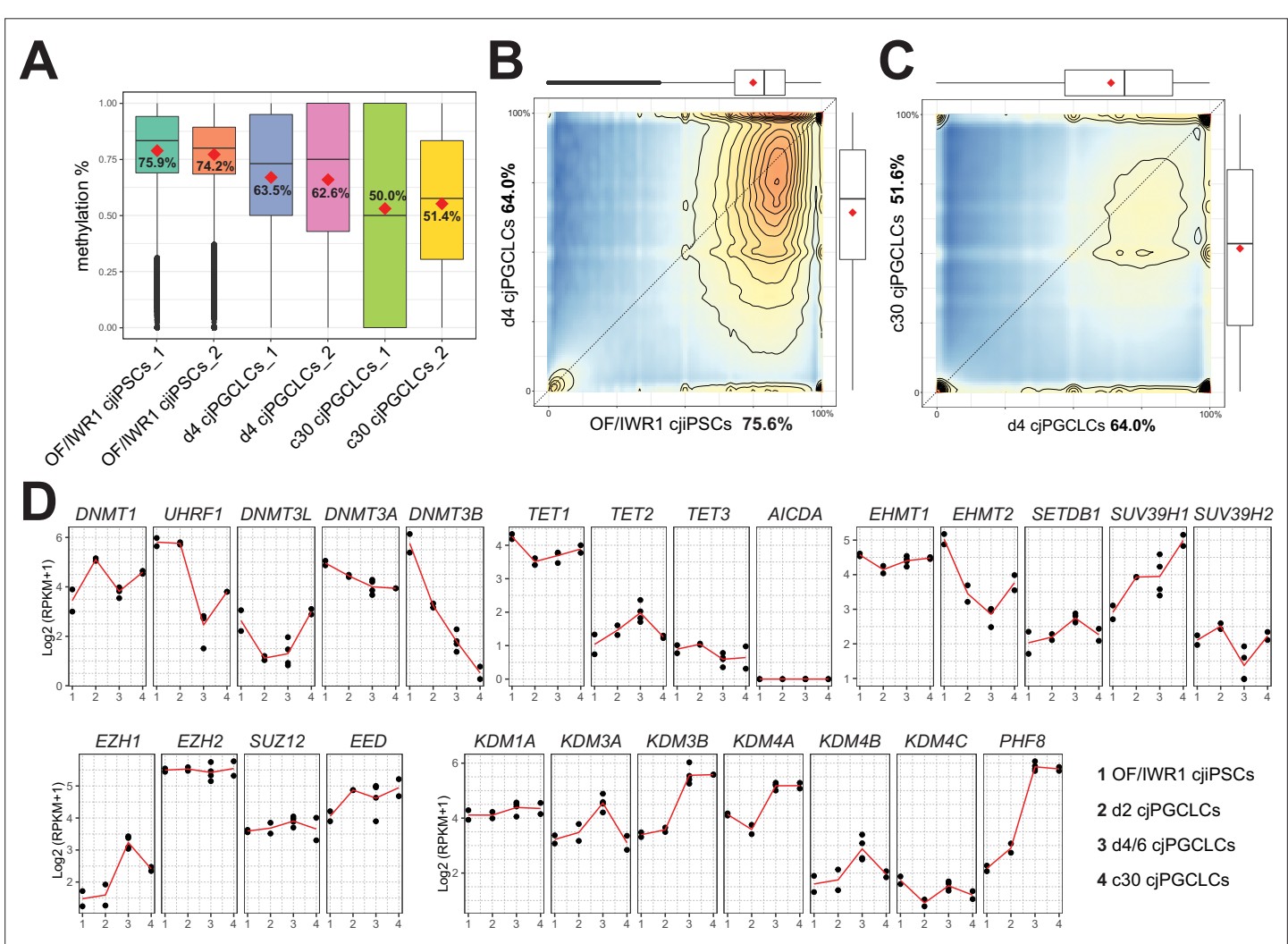

**Figure 8.** Genome-wide DNA methylation in *Callithrix jacchus* primordial germ cell-like cells (cjPGCLCs). (**A**) Boxplot showing overall CpG DNA methylation levels. Mean DNA methylation levels (red diamonds) are labeled. Center line, median; box limits, upper and lower quartiles; whiskers, 1.5× interquartile range. (**B, C**) DNA methylation levels of 2 kb tiles comparing genomes of OF/IWR1 cjiPSCs and d4 cjPGCLCs (**B**), or d4 PGCLCs and c30 cjPGCLCs (**C**). Mean methylation levels are labeled in the axis titles, and boxplots show the first and third quartiles and median methylation levels. (**D**) Gene expression dynamics during cjPGCLC induction and c30 expansion culture for genes associated with DNA methylation and histone modifications. Expression is shown as log$_2$(RPKM + 1).

*et al., 2015*). We extended this study and further provided the first comprehensive immunophenotypic and transcriptomic profile of cjPGCs from E50 embryos (CS11) (*Figures 1 and 2*). We discovered that cjPGCs displayed immunophenotypic and transcriptomic features characteristic of endogenous PGCs of humans and old-world monkeys. For example, they expressed key primate germ cell specifier genes (*SOX17, SOX15, TFAP2C PRDM1, PRDM14* [at low levels]), and lacked *SOX2*. In contrast, mouse germ cells highly express *SOX2* but only transiently express *SOX17* immediately after specification (*Kurimoto et al., 2008*; *Western et al., 2005*). Recent studies suggest that these features are also shared in rabbits and pigs, suggesting that the germline gene regulatory networks functioning in primates are more widespread evolutionary than that of rodents (*Kobayashi et al., 2021*; *Zhu et al., 2021*). Importantly, cjPGCs in E50 (CS11) embryos are primarily pre-migratory (i.e., localized within the hindgut endoderm), and exhibit features of early PGCs (i.e., lack of DDX4 or DAZL expression) similar to PGCs of cynomolgus monkeys at the corresponding developmental stage (*Figures 1E and 2E*; *Sasaki et al., 2016*). Interestingly, human/cynomolgus PGCs at the same chronologic age (E50) already colonize the gonads and upregulate DDX4/DAZL (*Sasaki et al., 2016*; *Castrillon et al., 2000*). This finding is likely due to the overall delay in early post-implantation embryo development in marmoset and suggest that germ cell development is synchronized with overall embryo development rather than chronologic age (*Phillips, 1976*).

With immunophenotypic and transcriptomic characterization of cjPGCs in hand, we were now able to validate methods required to generate cjiPSCs, which could subsequently be used to assess molecular events associated with germline induction. To this end, we first established various culture methods for cjiPSCs (*Figure 3—figure supplement 1*, *Figure 3—figure supplement 2*). Previous studies suggested that cynomolgus ESCs cultured in DK20 on MEF were prone to differentiate, but that inhibition of WNT signaling in these cultures stabilized the undifferentiated state (*Sakai et al., 2020*). Consistently, we found that inhibition of WNT signaling by IWR1 stabilized the undifferentiated state of cjiPSC cultured on MEF (*Figure 3—figure supplement 2*). In contrast, our newly established PluriSTEM-based FF cjiPSC culture method facilitated stable maintenance of an undifferentiated state, regardless of the presence or absence of IWR1 (*Figure 3—figure supplement 1C*, *Figure 3—figure supplement 2D*, *Figure 6—figure supplement 1B, C*). This might be due to the inclusion of proprietary factors in the PluriSTEM base medium that support the undifferentiated state. Nonetheless, despite the distinct propensities toward differentiation, the cjiPSCs used in this study exhibited gene expression characteristics of primed-state pluripotency and could be maintained across multiple passages in all of above culture conditions.

Our successful identification of culture conditions capable of generating and maintaining cjiPSCs allowed us to next compare their competency to differentiate into cjPGCLCs. Although cjiPSCs cultured under various conditions were all pluripotent as evidenced by pluripotency-associated gene expression and outcomes of a trilineage differentiation assay, they differed substantially in germline competency. Specifically, we noted that FF cjiPSCs (with or without IWR1) had no germline competency whereas OF and OF/IWR1 cjiPSCs had modest and high germline competency, respectively (*Figure 3*, *Figure 3—figure supplement 3*). Transcriptomes of FF, OF, OF/IWR1 cjiPSCs aligned accordingly on the PC space, where FF cjiPSCs were the most distant from and OF/IWR1 cjiPSCs were the closest to cjPGCLCs. As both FF and OF/IWR1 cjiPSCs are pluripotent and can be cultured stably without overt meso/endodermal differentiation (*Figure 3—figure supplement 2D*), these findings suggest that pluripotency/differentiation state per se might not be the primary reason why FF cjiPSCs are completely devoid of germline competency. Rather, a comparison of OF/IWR1 (vs. OF) and OF (vs. FF cjiPSCs) revealed that upregulation of a number of genes related to UPS protein catabolism, particularly those within the E3 ubiquitin ligase complex known as the Skp, Cullin, F-box containing (SCF) complex (e.g., *UBE3A*, *FBXW7*), correlated with increased germline competency (*Figure 6—figure supplement 1D*). The vast majority of these genes continued to be expressed in d2 cjPGCLCs, which suggests their potential role in germ cell specification (*Figure 6—figure supplement 1E, F*). In line with this, recent studies highlighted the critical role of the UPS system, and in particular FBXW7, in regulation of pluripotency and germ cell development (*Buckley et al., 2012*; *Zhou et al., 2020*; *Kanatsu-Shinohara et al., 2014*). Further mechanistic studies investigating the role of UPS and protein catabolism in germline competency and cjPGCLCs specification are warranted.

In this study, we provide evidence that cjPGCLCs can be derived from cjiPSCs through direct floating culture of OF/IWR1 cjiPSCs in the presence of a PGCLC induction cocktail (*Figure 3*; *Sasaki*

*et al., 2015*; *Sakai et al., 2020*). Under these conditions, cjPGCLCs were induced in a highly efficient manner, with the number of PDPN+ cjPGCLCs peaking at d4 (~600 cells/aggregate). The marmoset germline induction efficiency is higher than that of hPGCLCs induced under direct floating culture and similar to those induced through a stepwise method (2D induction of incipient mesoderm-like cells [iMeLCs] by ACTIVIN A and WNT agonist, CHIR99021, followed by floating culture with PGCLC induction factors) (*Figure 3*; *Sasaki et al., 2015*). Since the iMeLC induction step is not essential for robust PGCLC induction in cynomolgus monkeys, these results highlight differential requirements for WNT/NODAL/ACTIVIN signaling to prime PGCLC specification, and/or differential endogenous production by the aggregates themselves (*Sakai et al., 2020*). Nevertheless, lineage trajectory analysis of emerging cjPGCLCs revealed the transient upregulation of mesodermal program (e.g., *EOMES*, *MIXL1*) at d2 cjPGCLCs (*Figures 6D, F, 7F and G*), similar to d2 cynomolgus PGCLCs and human iMeLCs (*Sasaki et al., 2015*; *Kojima et al., 2017*; *Sakai et al., 2020*), suggesting that germline developmental programs from pluripotency are overall conserved across primate species.

Interestingly, a previous study by Okano and colleagues failed to produce cjPGCLCs from cjiPSCs by floating culture using a PGCLC induction cocktail similar to ours (*Yoshimatsu et al., 2021*). This may in part be due to the relatively poor germline competency of cjiPSCs/ESCs used in that study, which were cultured under conventional OF condition without IWR1. To overcome the lack of induction, these authors employed an alternative approach in which cytokine-based induction was combined with over-expression of key PGC specifier transcription factors, *PRDM1* and *SOX17*. Although this approach allowed induction of PRDM1-Venus+ cjPGCLCs from both cjESCs and cjiPSCs, the efficiency was variable, with two cjESC lines showing 30–40% induction and a cjiPSC line showing only 1.7% efficiency. Moreover, DDX4 was upregulated in some of PRDM1::Venus+ cells as early as d9-10. In primates including marmoset, DDX4 is not expressed in pre-migratory PGCs in vivo (*Figures 1E and 2E*; *Sasaki et al., 2016*), and human PGCLCs upregulate DDX4 only after prolonged xrTestis/xrOvary culture in vitro (*Hwang et al., 2020*; *Yamashiro et al., 2018*), suggesting that the induction method utilized in this study might not fully recapitulate the physiological germ cell developmental trajectory. Whether over-expression of transcription factors can drive cjPGCLC formation from the OF/IWR1 cjiPSCs with high germ cell competency that we established in this study remains to be determined.

In summary, the in vitro platform described here enables efficient induction of cjPGCLCs from cjiPSCs, which will serve as a foundation for analyzing mechanisms of PGC specification in marmoset monkeys. Although the road ahead will likely be long, efforts to develop IVG in marmosets, which allows functional validation by fertilization and creation of offspring, may ultimately provide a suitable preclinical model of human IVG.

## Materials and methods
### Collection of marmoset embryo samples

Marmosets were housed at the Southwest National Primate Research Center (SNPRC), Texas Biomedical Research Institute, an AAALAC accredited institution. All procedures were reviewed and approved by the Texas Biomedical Research Institute IACUC (1772CJ). Marmosets at the SNPRC were maintained under standardized husbandry conditions as described previously (*Layne and Power, 2003*). For breeding, marmosets were housed in male-female monogamous pairs. Females received an unsedated transabdominal ultrasound monthly until pregnancy was confirmed with a GE Logiq portable ultrasound machine. Females were habituated to manual restraint and received positive reinforcement during the procedure. After a pregnancy was detected (<30 days estimated gestational age), pregnancy progression was assessed every 14 days. The gestational age of embryos was estimated with crown-rump length, assessed via ultrasound, which has previously been found to reliably estimate gestational age in marmosets to within ±3 days (*Jaquish et al., 1995*; *Tardif et al., 1998*).

Embryos at E50 were recovered from the uteri obtained through hysterectomy performed under full anesthesia. First, the endometrium was exposed by dissection of the serosa and myometrium at the lateral side of the explanted uterus. Then, the exposed endometrium was carefully opened along the cervix-to-fundus direction to approach the uterine cavity, from which embryonic sacs were recovered and collected into dishes containing RPMI 1640 medium. Three embryos (CS11) were isolated from embryonic sacs and photographed. After removal of the amnion and yolk sac, the posterior portions of the embryos were dissected and used in histologic analysis or single-cell RNA-seq.

## Marmoset peripheral blood mononuclear cell collection and reprogramming to cjiPSCs

Marmoset whole blood was collected into Na-heparin vacuum tubes, mixed with an equal volume of wash buffer (phosphate buffered saline [PBS] containing 2% fetal calf serum [FCS]) and layered onto Lymphoprep density separation medium in Sep-Mate tubes (both from STEMCELL Technologies). Cells were spun at 1200 g × 20 min, and the layer containing the PBMCs was collected in a separate tube. Isolated PBMCs were washed twice with wash buffer, with centrifugation at 300×g for 8 min. Isolated PBMCs were counted and cryopreserved in FCS with 10% dimethyl sulfoxide in a Mr Frosty freezing chamber, first at –80°C overnight and then for long-term storage in a –150°C cryogenic freezer. Marmoset PBMCs were reprogrammed with a CytoTune-iPS 2.0 Sendai Reprogramming Kit (Thermo Fisher) according to the manufacturer's directions. Briefly, PBMCs were cultured in StemPro-34 medium containing 100 ng/ml SCF, 100 ng/ml FLT-3, 20 ng/ml IL-3, and 20 ng/ml IL-6 (PBMC-Medium) for 4 days. On the fourth day (day 0) Sendai viruses (KOS, C-Myc, and KLF-4) were added to the PBMCs in a 5:5:3 ratio, and the cells were cultured until day 3 in PBMC-Medium. On day 3, the cells were plated onto a mouse embryonic fibroblast (MEF) feeder layer (23,400 cells/cm$^2$, pretreated with Mitomycin C [Sigma, M0503] at 10 μg/ml for 2 hr at 37°C, 5% $CO_2$) at a concentration of 50,000–500,000 PBMCs per well of a six-well plate in StemPro-34 medium without cytokines. The medium was changed daily until day 8, at which point the medium was changed to Pluristem medium (MilliporeSigma). Between 14 and 28 days, individual colonies formed, and each individual colony was handpicked and transferred clonally to a new well containing MEFs. These cjiPSCs were passaged by mechanical dissection of colonies into clumps until cryopreservation in freezing media (90% FBS and 10% DMSO).

Quantitative reverse transcription PCR was used to determine whether the cjiPSCs had cleared the Sendai virus reprogramming factors. cjiPSCs were collected as described above for passaging and pelleted. The cell pellet was resuspended in 0.5 ml TRIzol, and total RNA was isolated with a Direct-zol RNA MiniPrep kit (Zymo Research). Two separate assays were performed to ensure that the cjiPSCs were free of mycoplasma contamination. First, the cjiPSC colonies were stained with DAPI to assess the presence of extranuclear DNA characteristic of mycoplasma infection. Second, cjiPSCs were harvested with 0.5 mM EDTA in PBS and pelleted at 400 g for 5 min. Genomic DNA was isolated from the pellet with a QIAamp DNA mini kit (Qiagen). Genomic DNA was screened with a LookOut Mycoplasma PCR Detection Kit (MilliporeSigma) according to the manufacturer's instructions. Only samples and that showed a positive control band after PCR and did not show the mycoplasma band were considered negative. G-band karyotype analyses were performed with Cell Line Genetics (Madison, WI, USA).

## Culture of cjiPSCs

For feeder-free cjiPSC culture, the cjiPSCs (C6 and C10) were cultured on xeno-free recombinant Laminin-511 E8 fragment-coated dishes (TAKARA, iMatrix-511silk) with PluriSTEM Human ES/iPS cell media (Sigma-Aldrich, SCM130). The cells were passaged approximately every 6–7 days as clumps after treatment with 0.5 mM EDTA in PBS for 10 min. For OF culture, the cjiPSCs were cultured with DK20F20 (DMEM/F12 [Thermo Fisher, 11320-033] supplemented with 20% (vol/vol) KSR, 1 mM sodium pyruvate [Thermo Fisher, 11360-070], 2 mM GlutaMax [Thermo Fisher, 35050061], 0.1 mM NEAA, 0.1 mM 2-mercaptoethanol [Thermo Fisher, 21985-023], penicillin-streptomycin at 25 U/ml [Thermo Fisher, 15070063], and recombinant human bFGF) at 20 ng/ml on Mitomycin C-treated MEFs (2.5×10$^5$ cells/well of a six-well plate). For OF/IWR1 culture, IWR1 was added at 2.5 μM (Sigma, I0161). For single-cell passage, cells were dissociated into single-cell suspension every 6–7 days with Accutase (Sigma-Aldrich, A6964) and seeded at a density of 1×10$^5$ cells/9 cm$^2$. Culture medium was supplemented with 10 μM ROCK inhibitor (Tocris, 1254) until 24 hr after passage.

## Trilineage differentiation assay

Generation of embryoid bodies and trilineage differentiation were performed as described previously with minor modification (*International Stem Cell Initiative, 2018*). Briefly, cells were trypsinized, counted, and re-seeded in low-binding 96-well V-bottom plates at a density of 3000 cells/well in STEMdiff APEL medium (Stem Cell Technologies) supplemented with growth factors for each germ-layer differentiation: ectoderm (3 μM dorsomorphin, 3 μM SB431542, and 100 ng/ml FGF2), endoderm

(100 ng/ml Activin A, 1 ng/ml BMP4), and mesoderm (20 ng/ml Activin A, 20 ng/ml BMP4) differentiation. Embryoid bodies cultured in each condition were harvested at day 10 for qPCR analyses.

## Exome sequencing of marmoset DNA

We performed exome sequencing of genomic DNA isolated from cjiPSC lines (20201_6, 20201_7, 20201_10), the PBMC donor of these cjiPSC lines (38189), his sibling (38574), and twin pairs from two unrelated pregnancies (38668/38667 and 38922/38921). The animal genomic DNA was obtained from hair follicles. Isolated gDNA (10 ng) was subjected to exome selection with the Human xGen Exome Hyb Panel v.2 (IDT) probe set, essentially as previously described (*Chan et al., 2021*), and Nextera XT libraries were sequenced with paired-end 150 NovaSeq chemistry (Illumina) targeting 20× coverage. Reads were aligned to the v.3.2.1 of the *C. jacchus* genome with BWA-MEM v.0.7.17 (*Li, 2013*), and called SNP variants with GATK v.4.2.6.1 (*DePristo et al., 2011*). GATK best practices were used with minor alterations. Calling of genetic variants in marmosets has been performed only on a small scale, thus providing limited information for recalibration of base qualities. Because we found a substantial decline in base quality scores after recalibration with available data, we omitted BQSR. The average coverage across exons was 12.6–18.5×. After variant quality score recalibration, 26,171 biallelic SNPs were retained, with a minimum of 20× coverage in each sample. Relatedness was estimated using pairwise allele sharing across all sites. The chimeric fraction was estimated according to previously published statistical approaches (*Worley, 2014*). The within-sample allele frequency across all sites was indicative of the level of chimerism present. For example, at sites where the two individuals composing a sample carried alternative alleles, the chimeric fraction was the frequency of the chimeric allele such that the chimeric fraction was estimated from the distribution of within-sample allele frequencies. In a diploid individual, unfixed sites should display a sharp peak in allele frequency at approximately 50% after exclusion of homozygous sites. This distribution can shift according to the level of chimerism present. We devised a simple model wherein the chimeric fraction was estimated by maximum likelihood.

## cjPGCLC induction

cjiPSCs were maintained by feeder-free condition. These cells were harvested and subsequently cultured on Mitomycin C-treated MEFs in the presence of DK20F20 medium until it becomes confluent (6–7 days). In some experiments, IWR1 (2.5 µM) was added to DK20F20 medium. After treatment with Accutase, single-cell suspension were prepared for cjPGCLCs induction. The cjPGCLCs were induced by plating of 3500 cjiPSCs per well of a low cell binding V-bottom 96-well plate (Greiner, 651970) in GK15 (GMEM (Thermo Fisher, 11710035), 15% KSR, 0.1 mM NEAA, 2 mM L-glutamine, 1 mM sodium pyruvate, 0.1 mM 2-mercaptoethanol, and 25 U/ml penicillin-streptomycin) or aRB27 (Advanced RPMI 1640 [Thermo Fisher, 12633-012]), 2×B27 [Thermo Fisher, 17504044, 0.1 mM NEAA, 2 mM L-glutamine, and 25 U/ml penicillin-streptomycin] supplemented with 200 ng/ml of BMP4 (R&D Systems, 314 BP-010), human LIF at 1000 U/ml, SCF (R&D Systems, 255-SC-010) at 200 ng/ml, EGF (R&D Systems, 236-EG) at 100 ng/ml, and 10 mM ROCK inhibitor (Y-27632). The floating aggregates were cultured for as many as 8 days without replacement of the medium.

## cjPGCLC expansion culture

The cjPGCLC expansion culture was as described previously with minor modifications (*Murase et al., 2020*). Briefly, the STO cell line (American Type Culture Collection, 1503) was maintained in DMEM (Gibco, 11965-084) containing 10% FBS (Gibco) and penicillin-streptomycin at 25 U/ml. STO cells were treated with Mitomycin C (MMC) (Sigma, M0503) at 10 µg/ml for 2 hr and then harvested by trypsinization. Day 6 cjPGCLCs were cultured on STO cells treated with MMC in DMEM (Gibco, 11054-001) containing 15% KSR, 2.5% FBS, 0.1 mM NEAA, 2 mM L-glutamine, 0.1 mM 2-mercaptoethanol, and penicillin-streptomycin at 25 U/ml supplemented with 10 µM forskolin, SCF at 200 ng/ml, and bFGF at 20 ng/ml, and passaged every 10 days after sorting of PDPN$^+$ITGA6$^+$ cells with a FACSAria Fusion flow cytometer (BD Biosciences). We plated $1.0×10^4$ cells/well of 24-well plate on the day of passage, including 0.5 ml of medium supplemented with 10 µM Y27632, and added 0.5 ml of medium without Y-27632 on the next day. From the third day onward, the entire medium was replaced with 0.5 ml of fresh medium every 2 days.

## Generation of xenogeneic reconstituted testes

xrTestes were generated by aggregating FACS-sorted PDPN+ c10 cjPGCLCs with mouse fetal testicular somatic cells by using the following method (*Hwang et al., 2020*). First, to isolate fetal testicular somatic cells, E12.5 mouse embryos were isolated from timed pregnant ICR females and collected in chilled DMEM (Gibco) containing 10% FBS (Gibco) and 100 U/ml penicillin/streptomycin (Gibco). Fetal testes were identified by their appearance, and the mesonephros were removed by tungsten needles. Isolated testes were washed with PBS and then incubated with dissociation buffer for 15 min at 37°C with periodic pipetting. The dissociation buffer contained 1 mg/ml Hyaluronidase Type IV (Sigma), 5 U Dispase (Corning), and 5 U DNase (Qiagen) in wash buffer (100 U/ml penicillin/streptomycin and 0.1% BSA in DMEM/F12). After another PBS wash, testes were dissociated into single cells using 0.05% Trypsin-EDTA in PBS for 10 min at 37°C followed by quenching with FBS. Cell suspensions were strained through a 70 μm nylon cell strainer and centrifuged. The remaining cell pellet was then resuspended with MACS buffer (PBS containing 0.5% BSA and 2 mM EDTA) and incubated with anti-SSEA1 antibody MicroBeads (Miltenyi Biotec) for 20 min on ice before being washed with MACS buffer and centrifuged. The cell pellet was again resuspended in MACS buffer and then applied to an MS column (Miltenyi Biotec) according to the manufacturer's protocol. The flow-through cells were centrifuged, resuspended with Cell Banker Type I, and cryopreserved in liquid nitrogen until use. All centrifugations were performed at 232×*g* for 5 min and were followed by removal of the supernatant. To generate floating aggregates, c10 cjPGCLCs (5000 cells per xrTestis) and thawed fetal testicular somatic cells (60,000 cells per xrTestis) were mixed and plated in a Lipidure-coated U-bottom 96-well plate (Thermo Fisher Scientific, 174925) in Minimum Essential Medium alpha (α-MEM, Invitrogen) containing 10% KSR (Gibco), 55 μM 2-mercaptoethanol (Gibco), 100 U/ml penicillin/streptomycin (Gibco), and 10 μM Y-27632. After 2 days of floating culture, floating aggregates were transferred onto Transwell-COL membrane inserts (Corning, 3496) using a glass capillary. Membrane inserts were soaked in α-MEM supplemented as described above, without Y-27632. xrTestes were cultured at 37°C under an atmosphere of 5% $CO_2$ in air and one-half the volume of medium was changed every 3 days.

## IF analysis

For IF analysis, floating aggregates during cjPGCLC induction and xrTestes were fixed with 2% paraformaldehyde (Sigma) in PBS for 3 hr on ice, washed three times with PBS containing 0.2% Tween-20 (PBST) and then successively immersed in 10% and 30% sucrose (Thermo Fisher Scientific) in PBS overnight at 4°C. The fixed tissues were embedded in OCT compound (Thermo Fisher Scientific), frozen and sectioned to 10 μm thickness with a −20°C cryostat (Leica, CM1800). Sections were placed on Superfrost Microscope glass slides (Thermo Fisher Scientific), which were then air-dried and stored at −80°C until use. Before staining, slides were washed three times with PBS and then incubated with blocking solution (5% normal goat serum in PBST) for 1 hr. Slides were subsequently incubated with primary antibodies in blocking solution for 1 hr, then with secondary antibodies and 1 μg/ml DAPI in blocking solution for 50 min. Both incubations were performed at room temperature and followed by four PBS washes. For negative control of IF studies, samples were not treated with primary antibodies whereas the remaining procedures remained the same (please refer to *Figure 1—source data 1*). Slides were mounted in Vectashield mounting medium (Vector Laboratories) for confocal laser scanning microscopy analysis (Leica, SP5-FLIM inverted). Confocal images were processed with LeicaLasX (v.3.7.2).

For IF analyses, embryo samples were fixed in 10% buffered formalin (Fisher Healthcare) with gentle rocking overnight at room temperature. After dehydration, tissues were embedded in paraffin, serially sectioned at 4 μm thickness with a microtome (Thermo Scientific Microm HM325) and placed on Superfrost Microscope glass slides. Paraffin sections were then de-paraffinized with xylene. Antigens were retrieved by treatment of sections with HistoVT One (Nacalai USA) for 35 min at 90°C and then for 15 min at room temperature. The staining and incubation procedure for paraffin sections was similar to that for frozen sections, with the following modifications: the blocking solution was 5% normal donkey serum in PBST; the primary antibody incubation was performed overnight at 4°C; and slides were washed with PBS six times after each incubation. Slides were mounted in Vectashield mounting medium for confocal microscopic analysis.

For IF of expansion cultured cjPGCLCs, the cells were cultured on STO plated Glass Bottom Dishes (Matsunami, D11130H). At d10 of culture, the cells were fixed in 4% paraformaldehyde in PBS for

15 min at room temperature, washed three times with PBS for 5 min each, and incubated in 0.2% Triton X-100 in PBS for 10 min at room temperature. The cells were subsequently incubated with primary antibodies in blocking solution for 1 hr, then with secondary antibodies and 1 µg/ml DAPI in blocking solution for 50 min. Both incubations were performed at room temperature and were followed by four PBS washes. Images were captured and processed with confocal laser scanning microscopy.

## Fluorescence-activated cell sorting

Samples of d4, d6, and d8 cjPGCLCs, and expansion cultured cjPGCLCs were analyzed with FACS. Floating aggregates containing cjPGCLCs were dissociated into single cells with 0.1% trypsin/EDTA treatment for 15 min at 37°C with periodic pipetting. After the reaction was quenched by addition of an equal volume of FBS, cells were resuspended in FACS buffer (0.1% BSA in PBS) and strained through a 70 µm nylon cell strainer (Thermo Fisher Scientific) to remove cell clumps. For cjPGCLCs, ITGA6 weakly positive and PDPN positive fractions were sorted with a FACSAria Fusion flow cytometer (BD Biosciences). For expansion cultured cjPGCLCs, ITGA6 positive and PDPN positive fractions were sorted with the FACSAria Fusion instrument. All FACS data were collected in FACSDiva Software v.8.0.2 (BD Biosciences). For analysis/sorting of cjPGCLCs with cell-surface markers, cells dissociated with trypsin-EDTA/PBS were stained with fluorescence-conjugated antibodies for 15 min at room temperature. After cells were washed twice with FACS buffer, the cell suspension was filtered through a cell strainer and analyzed or sorted with a flow cytometer.

## qPCR analysis

FACS-sorted in vitro cells (cjPGCLCs and expansion cultured cjPGCLCs) were collected in CELLOTION (Amsbio). Embryoid bodies for trilineage differentiation assay, cjiPSCs and day 2 cjPGCLCs were collected in PBS, without FACS. Total RNAs were extracted from the cells with RNeasy Micro kits (QIAGEN, 74104) according to the manufacturer's instructions. The cDNA synthesis from 1 ng of total RNAs and the amplification of 3′ends were performed as described previously (*Nakamura et al., 2015*). The quality of the amplified cDNAs was validated on the basis of the Ct values determined by qPCR with the primers listed in *Supplementary file 1*. Quantitative PCR was performed with Power SYBR Green PCR Master Mix (Thermo Fisher, 4367659) and a StepOnePlus real-time qPCR system (Applied Biosystems) according to the manufacturer's instructions.

## 10× Genomics single-cell RNA-seq library preparation

The posterior portions of CS11 embryos were dissected, rinsed with PBS twice, and dissociated into single cells with 0.1% trypsin/EDTA treatment for 15 min at 37°C with periodic pipetting. After the reaction was quenched by addition of an equal volume of FBS, then strained through a 70 µm nylon cell strainer, cells were resuspended in FACS buffer (0.1% BSA in PBS). Cells were loaded into chromium microfluidic chips with the Chromium Next GEM Single Cell 3′ Reagent Kit (v.3.1 chemistry) and then used to generate single-cell gel bead emulsions (GEMs) with the Chromium Controller (10× Genomics) according to the manufacturer's protocol. GEM-RT was performed in a C1000 Touch Thermal Cycler with 96-Deep Well Reaction Module (Bio-Rad). All subsequent cDNA amplification and library construction steps were performed according to the manufacturer's protocol. Libraries were sequenced with a 2×150 paired-end sequencing protocol on an Illumina HiSeq 4000 or NovaSeq 6000 instrument.

## Mapping reads of 10× Chromium scRNA-seq and data analysis

Raw data were demultiplexed with the mkfastq command in Cell Ranger (v.6.1.2) to generate Fastq files. Then raw reads were mapped to the *C. jacchus* (calJac4) reference genome from USCS. Raw gene counts were obtained with Cell Ranger.

Secondary data analyses were performed in R (v.4.1.0) with Seurat (v.4.1.1). UMI count tables were first loaded into R with the Read10X_h5 function, and Seurat objects were built from each sample. For characterization of cjPGCs, of 34,458 total cells captured in the library, we detected 2224–4760 median genes/cell at a mean sequencing depth of 46,964–102,934 reads/cell. For characterization of cjPGCLCs, of 9098 total cells captured in the libraries (three libraries comprising 572, 4292, and 4234 cells from cjiPSCs, d2 and d6 cjPGCLCs, respectively), we detected 5612 genes/cell at a mean sequencing depth of 44,379 reads/well. Samples were combined, and the effects of library size were

regressed out by SCTransform during normalization in Seurat and then converted to $\log_2$ (CP10 M+1) values. Cells were clustered with a shared nearest neighbor modularity optimization-based clustering algorithm in Seurat. Clusters were annotated on the basis of previously characterized marker gene expression with the FeaturePlot function and the gene expression matrix file, and cluster annotation was generated for downstream analyses. Dimensional reduction was performed with the top 3000 highly variable genes and the first 30 PCs with Seurat. DEGs in different clusters were calculated with Seurat findallmarkers, with average $\log_2$ fold change thresholds of above 0.25, p-value <0.01, and FDR <0.01. DEGs between two groups in the scatter plot were identified with edgeR 3.34.1 through a quasi-likelihood approach, with the fraction of detected genes per cell as the covariate. The DEGs were defined as the genes with FDR <0.01, p-value <0.01, and $\log_2$ fold change above 1. The cell cycle was analyzed with CellCycleScoring in Seurat. Data were visualized with R (v.4.1.0). Genes in the heatmap were hierarchically clustered according to the Euclidean distance, scaled by row, and then visualized with pheatmap. Gene ontology enrichment was analyzed with DAVID v.6.8.

For analyzing RNA velocity, spliced, unspliced, and ambiguous counts tables were generated using STAR (version 2.7.10b) with parameters as `--soloType CB_UMI_Simple --soloBarcodeReadLength` 0 `--soloUMIlen` 12 `--soloStrand` Forward `--soloUMIdedup` 1 MM_CR `–soloCBmatchWLtype` 1 MM_multi_Nbase_pseudocounts `--soloUMIfiltering` MultiGeneUMI_CR `--soloCellFilter` EmptyDrops_CR `--clipAdapterType` CellRanger4 `--outFilterScoreMin` 30 `--soloFeatures` Gene GeneFull Velocyto. Then, the output results were imported into scVelo (v0.2.5) for further analysis. During velocity analysis, top 5000 variable genes were used for processing the data and dynamical mode was used to estimate the velocity. The remaining parameters were default. After estimation of velocity, velocity vector fields were projected to the same PHATE embedding.

Pseudotime and principal curve were analyzed by slingshot (v.2.6.0) using default parameters. PHATE components were used to predict pseudotime and fitting of principal curve using getLineages and getCurves with default setting. Then, the calculated pseudotime and principal curve were projected to the same PHATE embedding.

For cross-species comparison between human PGCLCs and cjPGCLCs, total of 12,405 genes conserved between human and marmoset are used. Human iPSC and human PGCLCs datasets are downloaded from previous study (GSE153819) (*Hwang et al., 2020*). Expression is log normalized by Seurat. DEGs are defined as $\log_2$ fold change above 0.5, p-value <0.05, and FDR <0.05.

## Bulk RNA-seq library preparation

CjiPSCs cultured in the presence or absence of feeders, and in the presence or absence of IWR1, were collected. To minimize the contamination with feeder cells, >30 colonies of cjiPSCs cultured on feeder layer were randomly picked under an inverted microscope and pooled before isolation of total RNA. Total RNA was extracted with an RNeasy Plus Micro Kit (#74034, QIAGEN). RNA-seq libraries were made using an SMRT-Seq HT plus kit (#R400748, Takara) according to the manufacturer's protocol. Briefly, total RNA was quantified with a Qubit instrument, and RNA integrity was verified with a TapeStation. Then, 1 ng RNA was used for cDNA conversion with a one-step first-strand cDNA synthesis and double-stranded cDNA amplification protocol. cDNA was purified with AMPxp beads, its concentration was measured with a Qubit, and its quality was verified with a TapeStation. Next, 2 ng cDNA was used for library construction. Libraries were dual indexed and pooled according at equal molecular concentrations. Subsequently, 100-base pair reads were sequenced on the Illumina NextSeq 2000 platform.

## Bulk RNA-seq data analysis

Raw fastq files were demultiplexed with bcl2fastq2 (v.2.20.0.422). Barcodes and adapters were removed with Trimmomatic (v.0.32). Fastq files were mapped to the *C. jacchus* (calJac4) reference genome with STAR (v.2.7.10a). The raw gene count table was generated with featurecounts, and weakly expressed genes were filtered with edgeR with the filterByExpr function with default parameters. Briefly, the raw counts were normalized to library size, and then genes with counts per million above 10 were included in downstream analysis. DEGs were analyzed with edgeR (v.3.36.0) with $\log_2$ fold change >1, p-value <0.05, and FDR <0.05. Reads per kilobase per million (RPKM) values were calculated in edgeR, and the gene length was obtained from the UCSC table browser. Downstream

data analyses and visualization were performed with R (v.4.1.0). Hierarchical clustering was performed with hclust in R (v.4.1.0).

## Bisulfite sequencing and analysis

For collection of cjiPSCs for methylome analyses, colonies were picked under a microscope, then collected into a 1.5 ml tube for lysis. PGCLC aggregates were digested with 400 µl 0.25% trypsin for 15 min. Then, 100 µl FBS was used to stop digestion, and the lysates were pipetted well to obtain a single-cell suspension. The dissociated cells were stained with APC-conjugated anti-human PDPN and BV421-conjugated anti-human/mouse CD49f (ITGA6), and then the PDPN+ITGA6weak+ fraction was sorted for methylome analyses.

Cells were collected and lysed in 50 mM Tris (pH 8.0), 10 mM EDTA, 0.5% SDS, and 100 µg/ml proteinase K. Then, crude DNA was used to build a sequencing library according to the protocol of the Pico Methyl-Seq Library Prep Kit (Zymo, #D5455). The libraries were sequenced on the Illumina 2200 platform. Raw fastq files were demultiplexed with bcl2fastq2 (v.2.20). Barcode and index trimming was performed with Trim Galore (v.0.6.5) as follows: trim_galore `--quality` 30 `--phred33` `--illumina` `--stringency` 1 `--cores` 4 -e 0.1 `--fastqc` `--clip_R1` 10 `--three_prime_clip_r1` 10 `--length` 20. The trimmed fastq files were then mapped to calJac4 from USCS with Bismark (v.0.22.3) as follows: bismark `--parallel` 4 `--genome_folder` \$REF `--non_directional` `--score_min` L,0,–0.6. CpG methylation was extracted and analyzed with methylKit (v.1.22.0). Covered CpG loci were included in the analysis. The genome was tiled in 2 kb windows, and DNA methylation levels were summarized with methyKit (v.1.22.0). Data were visualized with R (v.4.1.0).

## Acknowledgements

We thank L King for carefully reviewing the manuscript and providing insightful comments. We thank J Marty and D Layne-Colon at Texas Biomedical Research Institute for marmoset embryo preparation. We appreciate Comparative Pathology Core at the University of Pennsylvania School of Veterinary Medicine for making paraffin blocks and T Moriwaki at the Sasaki lab for sectioning of the paraffin blocks. We thank C Malekshahi and D Beiting at the Center for Host-Microbial Interactions at the University of Pennsylvania School of Veterinary Medicine for cDNA library preparation and sequencing for bulk RNA-seq and WGBS. We thank A Vaughan for fluorescence microscope imaging of cultured cells. We thank members of the Sasaki lab and members of the NIDA Brain Initiative transgenic marmoset consortium for the discussion of the study. This work was supported in part by NIH grants U01 DA054170 (KS, BH, CN, JM, CR), R01 HD090007 (BH) and P51 OD011133 (CR), the Open Philanthropy funds from Silicon Valley Community Foundation (2019-197906) and Good Ventures Foundation (10080664) to KS. Results were generated in part with help from the UTSA Genomics Core which receives support from NIH grant G12-MD007591, NSF grants DBI-1337513, DBI-2018408.

## Additional information

### Funding

| Funder | Grant reference number | Author |
|---|---|---|
| Open Philanthropy Project | 2019-197906 | Kotaro Sasaki |
| Open Philanthropy Project | 10080664 | Kotaro Sasaki |
| National Institute on Drug Abuse | U01DA054170 | Brian P Hermann<br>John R McCarrey<br>Christopher S Navara<br>Corinna N Ross<br>Kotaro Sasaki |
| Eunice Kennedy Shriver National Institute of Child Health and Human Development | R01HD090007 | Brian P Hermann |

| Funder | Grant reference number | Author |
|---|---|---|
| National Institute on Aging | P51OD011133 | Corinna N Ross |
| National Institute on Minority Health and Health Disparities | G12MD007591 | Sean Vargas |
| National Science Foundation | DBI-1337513 | Sean Vargas |
| National Science Foundation | DBI-2018408 | Sean Vargas |

The funders had no role in study design, data collection and interpretation, or the decision to submit the work for publication.

### Author contributions

Yasunari Seita, Data curation, Formal analysis, Investigation, Visualization, Writing – original draft; Keren Cheng, Resources, Data curation, Software, Formal analysis, Investigation, Visualization, Writing – original draft; John R McCarrey, Conceptualization, Supervision, Funding acquisition, Writing – review and editing; Nomesh Yadu, Sean Vargas, Resources, Investigation; Ian H Cheeseman, Software, Formal analysis, Supervision, Investigation, Writing – original draft; Alec Bagwell, Formal analysis, Visualization; Corinna N Ross, Resources, Supervision, Funding acquisition, Investigation, Project administration; Isamar Santana Toro, Li-hua Yen, Resources; Christopher S Navara, Resources, Supervision, Funding acquisition, Investigation; Brian P Hermann, Funding acquisition, Investigation, Writing – original draft; Kotaro Sasaki, Conceptualization, Resources, Formal analysis, Supervision, Funding acquisition, Validation, Investigation, Visualization, Writing – original draft, Writing – review and editing

### Author ORCIDs

Keren Cheng http://orcid.org/0000-0002-9617-3584
Kotaro Sasaki http://orcid.org/0000-0002-5604-2651

### Ethics

This study was performed in strict accordance with the recommendations in the Guide for the Care and Use of Laboratory Animals of the National Institutes of Health. All procedures were reviewed and approved by the Texas Biomedical Research Institute IACUC (1772CJ).

### Decision letter and Author response

Decision letter https://doi.org/10.7554/eLife.82263.sa1
Author response https://doi.org/10.7554/eLife.82263.sa2

---

# Additional files

### Supplementary files
- Supplementary file 1. Primers used in this study.
- MDAR checklist

### Data availability

Accession number for RNA-seq and whole genome bisulfite sequencing generated in this study is GSE210576. The Exome sequence data generated in this study are available as SRA BioProject PRJNA856282.

The following datasets were generated:

| Author(s) | Year | Dataset title | Dataset URL | Database and Identifier |
|---|---|---|---|---|
| Seita Y, Cheng K, McCarrey JR, Yadu N, Cheeseman I, Bagwell A, Vargas S, Navara C, Hermann BP, Sasaki K | 2023 | Efficient generation of marmoset primordial germ cell-like cells using induced pluripotent stem cells [scRNA-seq] | https://www.ncbi.nlm.nih.gov/geo/query/acc.cgi?acc=GSE209932 | NCBI Gene Expression Omnibus, GSE209932 |
| Seita Y, Cheng K, McCarrey JR, Yadu N, Cheeseman I, Bagwell A, Vargas S, Navara C, Hermann BP, Sasaki K | 2023 | Sequencing of Callithrix jacchus iPSC lines | http://www.ncbi.nlm.nih.gov/bioproject/?term=PRJNA856282 | NCBI BioProject, PRJNA856282 |

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

## Appendix 1

### Appendix 1—key resources table

| Reagent type (species) or resource | Designation | Source or reference | Identifiers | Additional information |
|---|---|---|---|---|
| Cell line (*Callithrix jacchus*) | 20201_6 (cjiPSC line) | This paper | | Reprogrammed from PBMCs (Donor ID: 38189) |
| Cell line (*Callithrix jacchus*) | 20201_7 (cjiPSC line) | This paper | | Reprogrammed from PBMCs (Donor ID: 38189) |
| Cell line (*Callithrix jacchus*) | 20201_10 (cjiPSCs line) | This paper | | Reprogrammed from PBMCs (Donor ID: 38189) |
| Cell line (*Mus musculus*) | STO | American Type Culture Collection | 1503 | Feeder for cjPGCLCs expansion |
| Biological sample (*Callithrix jacchus*) | 38189 | This paper | | Hair follicles from the donor 38189 (sibling of 38574) |
| Biological sample (*Callithrix jacchus*) | 38574 | This paper | | Hair follicles from the donor 38574 (sibling of 38189) |
| Biological sample (*Callithrix jacchus*) | 38921 | This paper | | Hair follicles from the donor 38921 (twin sibling of 38922) |
| Biological sample (*Callithrix jacchus*) | 38922 | This paper | | Hair follicles from the donor 38922 (twin sibling of 38921) |
| Biological sample (*Callithrix jacchus*) | 38667 | This paper | | Hair follicles from the donor 38667 (twin sibling of 38668) |
| Biological sample (*Callithrix jacchus*) | 38668 | This paper | | Hair follicles from the donor 38668 (twin sibling of 38667) |
| Biological sample (*Callithrix jacchus*) | cj embryos (E50, Carnegie stage 11) | This paper | | Triplet embryos freshly isolated by hysterectomy |
| Biological sample (*Callithrix jacchus*) | cj fetal testis | This paper | | Freshly isolated testis from cj fetus at gestational week 19 |
| Biological sample (*Callithrix jacchus*) | cj neonatal testis | This paper | | Freshly isolated testis from a neonate |
| Biological sample (*Mus musculus*) | Mouse fetal testicular somatic cells (E12.5) | This paper | | Used for xrTestis culture |
| Biological sample (*Mus musculus*) | Mouse embryonic fibroblast | This paper | | Isolated from E12.5 mouse embryo, used as feeder cells for cjiPSCs culture |
| Antibody | Anti-TFAP2C (mouse monoclonal) | Santa Cruz Biotechnology | sc-12762; RRID:AB_667770 | IF (1:200) |
| Antibody | Anti-SOX17 (goat polyclonal) | Neuromics | GT15094; RRID:AB_2195648 | IF (1:150) |
| Antibody | Anti-OCT3/4 (mouse monoclonal) | Santa Cruz Biotechnology | sc-5279; RRID:AB_628051 | IF (1:150) |
| Antibody | Anti-NANOG (goat polyclonal) | R&D Systems | AF1997; RRID:AB_355097 | IF (1:150) |
| Antibody | Anti-DDX4 (goat polyclonal) | R&D Systems | AF2030; RRID:AB_2277369 | IF (1:200) |
| Antibody | Anti-DAZL (rabbit polyclonal) | Abcam | ab34139; RRID:AB_731849 | IF (1:150) |
| Antibody | Anti-SOX2 (mouse monoclonal) | R&D Systems | MAB2018; RRID:AB_358009 | IF (1:150) |
| Antibody | Anti-LAMININ (rabbit polyclonal) | Abcam | ab11575; RRID:AB_298179 | IF (1:200) |
| Antibody | Anti-H3K9me2 (rabbit polyclonal) | MilliporeSigma | 07441; RRID:AB_11212297 | IF (1:150) |

*Appendix 1 Continued on next page*

*Appendix 1 Continued*

| Reagent type (species) or resource | Designation | Source or reference | Identifiers | Additional information |
|---|---|---|---|---|
| Antibody | Anti-H3K27me3 (rabbit polyclonal) | MilliporeSigma | 07449; RRID:AB_310624 | IF (1:150) |
| Antibody | Anti-SOX9 (rabbit polyclonal) | MilliporeSigma | AB5535; RRID:AB_2239761 | IF (1:150) |
| Antibody | Anti-PRDM1 (rabbit monoclonal) | Abcam | ab198287 | IF (1:100) |
| Antibody | Alexa Fluor 488-conjugated anti-rabbit IgG (donkey polyclonal) | Life Technologies | A21206; RRID:AB_2535792 | IF (1:500) |
| Antibody | Alexa Fluor 488-conjugated anti-mouse IgG (donkey polyclonal) | Life Technologies | A32766; RRID:AB_2762823 | IF (1:500) |
| Antibody | Alexa Fluor 568-conjugated anti-mouse IgG (donkey polyclonal) | Life Technologies | A10037; RRID:AB_2534013 | IF (1:500) |
| Antibody | Alexa Fluor 568-conjugated anti-rabbit IgG (donkey polyclonal) | Life Technologies | A10042; RRID:AB_2534017 | IF (1:500) |
| Antibody | Alexa Fluor 647-conjugated anti-goat IgG (donkey polyclonal) | Life Technologies | A21447; RRID:AB_2535864 | IF (1:500) |
| Antibody | Alexa Fluor 647-conjugated anti-rabbit IgG (donkey polyclonal) | Life Technologies | A31573; RRID:AB_2536183 | IF (1:500) |
| Antibody | Alexa Fluor 647-conjugated anti-human PDPN (rat monoclonal) | Biolegend | 337007; RRID:AB_1595538 | FACS (5 µl per test) |
| Antibody | BV421-conjugated anti-mouse CD49f (rat monoclonal) | Biolegend | 313623; RRID:AB_2562243 | FACS (5 µl per test) |
| Antibody | Alexa Fluor 488-conjugated anti-human/mouse SSEA3 (rat monoclonal) | Biolegend | 330305; RRID:AB_1279441 | FACS (5 µl per test) |
| Antibody | PE-conjugated anti-human SSEA4 (mouse monoclonal) | Biolegend | 330405; RRID:AB_1089207 | FACS (5 µl per test) |
| Antibody | Anti-SSEA1 antibody Microbeads (mouse monoclonal) | Miltenyi Biotec | 130-094-530; RRID:AB_2814656 | MACS removal of germ cells (20 µl per test) |
| Peptide, recombinant protein | Recombinant human BMP4 | R&D Systems | 314BP-010 | cjPGCLCs induction |
| Peptide, recombinant protein | Recombinant human LIF | MilliporeSigma | LIF1010 | cjPGCLCs induction |
| Peptide, recombinant protein | Recombinant human SCF | R&D Systems | 255-SC-010 | cjPGCLCs induction |
| Peptide, recombinant protein | Recombinant human EGF | R&D Systems | 236-EG | cjPGCLCs induction |

*Appendix 1 Continued on next page*

*Appendix 1 Continued*

| Reagent type (species) or resource | Designation | Source or reference | Identifiers | Additional information |
|---|---|---|---|---|
| Peptide, recombinant protein | Recombinant human/ murine/rat Activin-A | R&D Systems | 338-AC | Trilineage differentiation |
| Peptide, recombinant protein | Recombinant human FLT-3 ligand | Gibco | PHC9414 | Generation of cjiPSCs lines |
| Peptide, recombinant protein | Recombinant human basic FGF | Gibco | PHG0264 | Generation of cjiPSCs lines |
| Peptide, recombinant protein | Recombinant human IL-3 | Gibco | PHC0034 | Generation of cjiPSCs lines |
| Peptide, recombinant protein | Recombinant human IL-6 | Gibco | PHC0065 | Generation of cjiPSCs lines |
| Peptide, recombinant protein | Recombinant human SCF | Gibco | PHC2111 | Generation of cjiPSCs lines |
| Peptide, recombinant protein | iMatrix-511silk (Laminin-511 E8 fragment) | TAKARA | 892021 | Feeder free cjiPSCs culture |
| Chemical compound, drug | Mitomycin C | MilliporeSigma | M0503 | Inactivation of feeder cells |
| Chemical compound, drug | Y-27632 dihydrochloride (ROCK inhibitor) | Tocris | 1254 | cjPGCLCs induction |
| Chemical compound, drug | Dorsomorphin | MilliporeSigma | P5499-5MG | Trilineage differentiation |
| Chemical compound, drug | SB431542 | MilliporeSigma | S4317 | Trilineage differentiation |
| Commercial assay or kit | CytoTune-iPS 2.0 Sendai Reprogramming Kit | Thermo Fisher | A16517 | |
| Commercial assay or kit | LookOut Mycoplasma PCR Detection Kit | MilliporeSigma | MP0035-1KT | |
| Commercial assay or kit | Chromium Single Cell 3" GEM, Library & Gel Bead Kit v.3 | 10× Genomics | PN-1000092 | |
| Commercial assay or kit | SMRT-Seq HT plus kit | TAKARA | R400748 | |
| Commercial assay or kit | Pico Methyl-Seq Library Prep Kit | Zymo Research | D5455 | |
| Software, algorithm | FACSDiva software v.8.0.2 | BD Biosciences | | |
| Software, algorithm | Leica LasX c 3.7.2 | Leica | | |
| Software, algorithm | R v.4.1.0 | https://www.r-project.org | | |
| Software, algorithm | DAVID v.6.8 | https://david.ncifcrf.gov | | |

