## [Editor Report]

This nicely done paper describes a method for robust differentiation of the common marmoset induced pluripotent stem cells (iPSCs) into primordial germ cell-like cells and subsequently into spermatogonia-like cells when combined with testis somatic cells. The data suggest that marmosets are very similar to humans and macaques.

---

## [Decision Letter]

**Decision letter after peer review:**

Thank you for submitting your article "Efficient generation of marmoset primordial germ cell-like cells using induced pluripotent stem cells" for consideration by *eLife*. Your article has been reviewed by 2 peer reviewers, and the evaluation has been overseen by a Reviewing Editor and Marianne Bronner as the Senior Editor. The reviewers have opted to remain anonymous.

The reviewers have discussed their reviews with one another, and the Reviewing Editor has drafted this to help you prepare a revised submission. The full reviews are included below for further clarification

Essential revisions:

1. The functionality of generated cells (both germ cells and the starting iPSC populations) is critical.

2. A more detailed assessment of the transcriptional similarities and other features of in vitro-generated PGCs and prospermatogonia versus in vivo counterparts should be provided.

3. Clarification of differentiation methods and discussion of potential intermediate/primed cell states in the differentiation protocol need to be provided.

4. It would be optimal to show how the molecular identity of PGCLC changes in the presence of somatic cells in an unbiased manner.

*Reviewer #1 (Recommendations for the authors):*

To enhance the novelty of the work and demonstrate the functionality of the cells generated, it may be helpful to consider:

1) Which genes and pathways are differentially expressed between in vitro derived PGCLC and in vivo derived PGCs? Can these signaling pathways be reconstituted to push differentiation further in vitro? How accessible are earlier timepoints (before day 50) and do the in vitro derived PGCLC resemble a particular time-point in development or are the in vitro derived cells just slightly off from in vivo? Might velocity help infer whether these in vitro derived cells are truly progressing towards in vivo PGCLC, or these are different states or only look clustered because of the inclusion of IPSC which are distinct? Removing IPSc and comparing the germ cells directly may be informative.

2)The reconstituted gonad suggests the successful generation of spermatogonia-like cells. How do these cells compare to in vivo-derived cells? Furthermore, analysis of these early timepoints of the recombinant testis may allow the assessment of the maturation of the in vitro derived PGCLC compared to in vivo.

3) What percent of the spermatogonia-like cells in reconstituted testis express DAZL and/or DDX4? Flow plots with these markers may be informative.

4) Can these in vitro-derived spermatogonia reconstitute full spermatogenesis in vivo and give rise to pups?

5) Given the results of this paper it seems that marmosets, macaques, and humans have similar PGC genetic circuitry, diving deeper into a comparative analysis may reveal novel insights about the species-specific network.

*Reviewer #2 (Recommendations for the authors):*

The following points should be addressed:

1. It is noted that OF/IWR1 cjiPSCs are transcriptionally very similar to cjPJCLCs. This raises doubts as to whether these two cell types are really distinct, i.e., whether the cjPGCLCs generated have been derived from a true iPSC state. Are OF/IWR1 cjiPSCs an intermediate primed/differentiated state that is required for germline induction from iPSCs? How dissimilar are OF/IWR1 cjiPSCs and cjPGCLCs? Are OF/IWR1 cjiPSCs still pluripotent (see below)?

2. Comparisons between cjPGCs and cjPGCLCs are limited to Pearson's correlation analysis. A more detailed description of the transcriptomic similarities between these in vivo and in vitro cell types should be provided.

3. While the absence of mycoplasma and a stable karyotype are encouraging features when describing the iPSC lines that are generated, functional evidence of iPSC properties (pluripotency) is lacking. The authors should validate whether the iPSCs are pluripotent through tri-lineage differentiation using an embryoid body assay followed by either immunostaining or RT-PCR to detect germ layer markers. Please also provide expression results for pluripotency surface markers (likely to include SSEA3, 4 and PODXL for marmosets). OCT3/4, *SOX2*, and NANOG expression and morphology are not sufficient for iPSC validation. Please also compare validation results between iPSCs under each culture condition.

4. Negative controls should be included in IF results in the supplementary data. These should show both secondary antibody-only results, as well as staining in iPSCs as a further negative control.

5. Please provide a sentence in the introduction or the results to give context for why PDPN marks cjPGCLS, i.e., what does it mark and in which species?

6. The authors discuss methylation in cjPGCs and then later cjPGCLS. However, only UHRF1 and de novo methyltransferases are reported as associated with both cell types. Do cjPGCLCs show similar expression of demethylases and enzymes involved in the deposition of H3K9Me2/3 as reported for cjPGCs?

7. Spelling/grammatical errors: Pierson should be Pearson. in vitro/in vivo are not always italicised. Expension should be expansion in Figure 6.C. Callithrix Jacchus: the second word should not be capitalised in species names.

8. Results: Transcriptome accompanying formation of cjPGCLCs, paragraph 2: "Clustering analysis of variably expressed genes across the developmental trajectory…" It is very unclear which developmental trajectory is being referred to here. As far as can be seen, Cluster 1 refers to iPSCs, but which type is not specified; Cluster 2: Not specified, the authors simply state "along the trajectory"; which cell-line and which stage has produced this cluster must be reported. Cluster 3: 2D expansion cells, and Cluster 4: Refers to d2 cjPGCLCs. Taken together, these cell types do not represent a developmental trajectory. This needs to be clarified if indeed the authors intend to represent the trajectory from iPSCs to PGCLCs. A visual schematic would further clarify how these represent a trajectory.

9. The potential role of the Ubiquitin-Proteasome System (UPS) should only be described in the discussion. Evidence for this is not robust enough to be reported in the results, as this is based on one GO term, for "ubiquitin-dependent protein catabolic process".

10. Results displayed from whole genome bisulphite sequencing are quite limited. A more detailed analysis of this dataset should be provided.

11. The generation of DDX4 and DAZL positive cells in reconstituted testis culture that represent prospermatogonia are described as "a few". Please provide quantification to indicate the efficiency of generating these cells.

12. There appears to be a direct contradiction in the discussion, with a statement made in the results. The results state that the OF/IWR1 condition suppressed the problematic expression of genes involved in meso/endoderm differentiation. However, the discussion states that transcriptomic analysis revealed no differences in genes associated with meso/endodermal differentiation between iPSC culture conditions. Please clarify this.

13. Because IWR1 had no effect on iPSCs in FF conditions, the authors suggest factors in Pluristem better support the undifferentiated state. However, these cells are not germline competent. The best outcomes have been produced by OF conditions with IWR1. No explanation is provided for why these conditions support differentiation to the germ lineage. While this is attributed to the upregulation of genes associated with the UPS, no explanation is given for what factors in the media have caused this upregulation. An additional explanation is required here to provide a link between the growth factors and small molecules present in the superior condition, and the upregulation of these genes. Furthermore, the role of WNT inhibition in germline specification requires elucidation since it dramatically increases efficiency in the OF conditions. How might inhibition of this pathway interact with the OF conditions?

14. Figure 6: D: Please provide a legend for the clusters; E and F: It is not clear which plot represents which gene.

---

## [Author Response]

Essential revisions:1. The functionality of generated cells (both germ cells and the starting iPSC populations) is critical.

We thank the reviewers for their insightful comment. We understand the importance of functional assessment of in vitro derived germ cells, which in a strict sense, requires derivation of spermatozoa and fertilization to generate offspring. However, such a feat has not yet been accomplished using in vitro derived germ cells in any primate species. While we are currently attempting to overcome the multiple technical hurdles that challenge in vitro derivation of primate gametes and are setting up a transplantation scheme in which to assess the functionality of in vitro derived germ cells in our 5-year U01 consortium funded by NIH, it is not realistic to include such data in this revision in a timely manner. In regard to the functionality of cjiPSCs, we have added additional validation of cjiPSCs cultured under various conditions using flow cytometric evaluation of pluripotency markers (i.e., SSEA3, SSEA4) and a trilineage differentiation assay. These studies confirmed that both on feeder (with or without IWR1) and feeder free cjiPSCs bear pluripotency.

2. A more detailed assessment of the transcriptional similarities and other features of in vitro-generated PGCs and prospermatogonia versus in vivo counterparts should be provided.

In response to this comment, we have added extensive scRNA-seq-based characterization of cjPGCLCs in the revised manuscript (Figure 7A-M, Figure 7—figure supplement 1A-H). We have compared E50 cjPGCs in vivo and cjPGCLCs derived in vitro and demonstrated the overall similarities between E50 cjPGCs and d6 cjPGCLCs (Figure 7I-M). We also noted some differences between in vivo and in vitro cells, which might be exaggerated because of the limited number of cjPGCs available in this study (Figure 7I-M). We are currently undergoing extensive characterization of in vivo cjPGCs at various stages, which will enable more precise mapping of our cjPGCLCs in developmental coordinates in vivo. This will be published separately.

3. Clarification of differentiation methods and discussion of potential intermediate/primed cell states in the differentiation protocol need to be provided.

We have added more detailed description of cjPGCLCs induction and xrTestis generation in Materials and methods in our revised manuscript [page 28, Materials and methods, “Generation of xenogeneic reconstituted testes (xrTestes)]. Moreover, we have performed scRNA-seq to define an intermediate/primed cell state with transient upregulation of mesodermal program (Figure 7A-H).

4. It would be optimal to show how the molecular identity of PGCLC changes in the presence of somatic cells in an unbiased manner.

Although we are working towards comprehensively characterizing iPSC-derived germ cells beyond the PGCLC state (i.e., prospermatogonia), the induction efficiency is low and given the lack of fluorescence reporters/reliable surface markers, separating these cells from somatic cells to perform downstream analyses will be quite challenging. Given prior studies in other species published by us and other groups, inclusion of later stage gametes appears beyond the scope of a single manuscript. Thus, we will publish such studies in a separate paper in the near future.

Reviewer #1 (Recommendations for the authors):To enhance the novelty of the work and demonstrate the functionality of the cells generated, it may be helpful to consider:1) Which genes and pathways are differentially expressed between in vitro derived PGCLC and in vivo derived PGCs? Can these signaling pathways be reconstituted to push differentiation further in vitro? How accessible are earlier timepoints (before day 50) and do the in vitro derived PGCLC resemble a particular time-point in development or are the in vitro derived cells just slightly off from in vivo? Might velocity help infer whether these in vitro derived cells are truly progressing towards in vivo PGCLC, or these are different states or only look clustered because of the inclusion of IPSC which are distinct? Removing IPSc and comparing the germ cells directly may be informative.

In response to the comment, we have conducted scRNA-seq on cjPGCLCs, which allowed transcriptomic comparison with E50 cjPGCs in vivo in the same 10x platform. In particular, as the reviewer suggested, we have made the pairwise comparison between d6 cjPGCLCs and E50 cjPGCs. Key germ cell marker/specifier genes were equally expressed between cjPGCLCs in vitro and cjPGCs in vivo (Figure 7I-L). Neither cell types expressed post-migratory germ cell markers (e.g., *DDX4*, *DAZL*), suggesting that they are both at pre-migratory stage (Figure 1, 2, 3, 4). We also noted some transcriptional differences between E50 cjPGCs in vivo and d6 cjPGCLCs in vitro (Figure 7M). Some of these up- and down-regulated genes might be due to culture adaptation or differential nutritional environment (e.g., “actin cytoskeleton organization” or “response to insulin”). Notably, d6 cjPGCLCs showed more genes related to cell division/apoptosis, which might indicate that they are more proliferative. Some genes encoding zinc-finger proteins were more upregulated in E50 cjPGCs, which might reflect their differences of developmental stage or the absence of key signals in cjPGCLCs. These points have been addressed in the revised manuscript (page 13, Results, “scRNA-seq revealed lineage trajectory and gene expression dynamics during formation of cjPGCLCs”). Because we have sampled in vivo cjPGCs only at E50, it is difficult to map the exact developmental stage of cjPGCLCs on the in vivo germline trajectory. We are currently collecting cjPGCs at broad developmental stages, which will allow more precise determination of the developmental stage of cjPGCLCs in the near future.

2) The reconstituted gonad suggests the successful generation of spermatogonia-like cells. How do these cells compare to in vivo-derived cells? Furthermore, analysis of these early timepoints of the recombinant testis may allow the assessment of the maturation of the in vitro derived PGCLC compared to in vivo.

Although we are working towards comprehensively characterizing iPSC-derived germ cells beyond the PGCLC state (i.e., prospermatogonia), the induction efficiency is low and given the lack of fluorescence reporters/reliable surface markers, separating these cells from somatic cells to perform downstream analyses will be quite challenging. Given prior studies in other species published by us and other groups, inclusion of the detailed characterization of later stage gametes appears beyond the scope of a single manuscript. Thus, we will publish such studies in a separate paper in the near future.

3) What percent of the spermatogonia-like cells in reconstituted testis express DAZL and/or DDX4? Flow plots with these markers may be informative.

By IF analyses, we identified 0.89 and 3.3% of DAZL or DDX4 positive cells, respectively (DDX4^+^TFAP2C^+^ cells [4/123, 3.3% among all TFAP2C^+^ cells] and DAZL^+^TFAP2C^+^ cells [2/232, 0.86% among all TFAP2C^+^ cells]). Overall scarcity of cells and lack of fluorescence reporters (DAZL and DDX4 are cytoplasmic proteins necessitating technically challenging intracellular staining procedure to be assessed by flow cytometry), we were not able to provide the flow cytometric plots in this study. This point has been described in the revised manuscript (page 11, Results, “Maturation of cjPGCLCs into early prospermatogonia-like state”).

4) Can these in vitro-derived spermatogonia reconstitute full spermatogenesis in vivo and give rise to pups?

Please see response to Essential Revision point 1.

5) Given the results of this paper it seems that marmosets, macaques, and humans have similar PGC genetic circuitry, diving deeper into a comparative analysis may reveal novel insights about the species-specific network.

We have added cross-species comparison of transcriptomes performed in the same laboratory and using the same sequence chemistry (10x Chromium), which highlighted similarities and differences (Figure 7—figure supplement 2, described in page 14, Results, “scRNA-seq revealed lineage trajectory and gene expression dynamics during formation of cjPGCLCs” in the revised manuscript).

Reviewer #2 (Recommendations for the authors):The following points should be addressed:1. It is noted that OF/IWR1 cjiPSCs are transcriptionally very similar to cjPJCLCs. This raises doubts as to whether these two cell types are really distinct, i.e., whether the cjPGCLCs generated have been derived from a true iPSC state. Are OF/IWR1 cjiPSCs an intermediate primed/differentiated state that is required for germline induction from iPSCs? How dissimilar are OF/IWR1 cjiPSCs and cjPGCLCs? Are OF/IWR1 cjiPSCs still pluripotent (see below)?

Although OF/IWR1 cjiPSCs are closer to cjPGCLCs than cjiPSCs cultured in other conditions, they are pluripotent (as evidenced by trilineage differentiation assay, morphological assessment, and expression of pluripotency markers, Figure 3—figure supplement 2) and do not express most of key germ cell markers (Figure 6—figure supplement 1C). Our newly added scRNA-seq analyses also highlighted the differences between OF/IWR1 and cjPGCLCs and the molecular dynamics associated with the transition.

2. Comparisons between cjPGCs and cjPGCLCs are limited to Pearson's correlation analysis. A more detailed description of the transcriptomic similarities between these in vivo and in vitro cell types should be provided.

This has been now described using scRNA-seq data of both E50 cjPGCs and d6 cjPGCLCs.

3. While the absence of mycoplasma and a stable karyotype are encouraging features when describing the iPSC lines that are generated, functional evidence of iPSC properties (pluripotency) is lacking. The authors should validate whether the iPSCs are pluripotent through tri-lineage differentiation using an embryoid body assay followed by either immunostaining or RT-PCR to detect germ layer markers. Please also provide expression results for pluripotency surface markers (likely to include SSEA3, 4 and PODXL for marmosets). OCT3/4, SOX2, and NANOG expression and morphology are not sufficient for iPSC validation. Please also compare validation results between iPSCs under each culture condition.

We have performed tri-lineage differentiation assay for each culture condition and added surface marker expression data.

4. Negative controls should be included in IF results in the supplementary data. These should show both secondary antibody-only results, as well as staining in iPSCs as a further negative control.

We have included negative control data in the revised manuscript (Figure 1–Source Data 1).

5. Please provide a sentence in the introduction or the results to give context for why PDPN marks cjPGCLS, i.e., what does it mark and in which species?

We have added the following sentence: “As we and others have previously identified that TFAP2C, SOX17 and PDPN specifically mark pre-migratory/migratory PGCs and PGCLCs in humans and macaque monkeys^7,14–16^, we attempted to trace cjPGCs using these markers.” (page 5, Results, “Immunohistochemical characterization of pre-migratory cjPGCs”).

6. The authors discuss methylation in cjPGCs and then later cjPGCLS. However, only UHRF1 and de novo methyltransferases are reported as associated with both cell types. Do cjPGCLCs show similar expression of demethylases and enzymes involved in the deposition of H3K9Me2/3 as reported for cjPGCs?

We had already described demethylases and enzymes involved in the deposition of H3K9me2/3 for cjPGCLCs in Figure 8D (Figure 7D in the older version). Regarding demethylases, *TET1* was highly expressed in both cjPGCLCs and cjPGCs, which might partly explain the modest demethylation occurring in cjPGCLCs. In mice, *EHMT2* (~E7.5 onwards) followed by *EHMT1* (E9.5 onwards) were downregulated in PGCs, which seems to be the basis of low global H3K9Me2 levels(Sekl et al., 2007). Similar to human PGCLCs (Sasaki et al., 2015), we found that cjPGC/cjPGCLCs both show marked downregulation, which might explain the low global H3K9me2 levels. These points have been added to the revised manuscript (page 15, Results, “Global DNA methylation in cjPGCLCs”).

7. Spelling/grammatical errors: Pierson should be Pearson. in vitro/in vivo are not always italicised. Expension should be expansion in Figure 6.C. Callithrix Jacchus: the second word should not be capitalised in species names.

We are grateful that the reviewer pointed out these errors. These have been corrected.

8. Results: Transcriptome accompanying formation of cjPGCLCs, paragraph 2: "Clustering analysis of variably expressed genes across the developmental trajectory…" It is very unclear which developmental trajectory is being referred to here. As far as can be seen, Cluster 1 refers to iPSCs, but which type is not specified; Cluster 2: Not specified, the authors simply state "along the trajectory"; which cell-line and which stage has produced this cluster must be reported. Cluster 3: 2D expansion cells, and Cluster 4: Refers to d2 cjPGCLCs. Taken together, these cell types do not represent a developmental trajectory. This needs to be clarified if indeed the authors intend to represent the trajectory from iPSCs to PGCLCs. A visual schematic would further clarify how these represent a trajectory.

Our original analysis revealed which cell lines produced clusters (Figure 6A) and defined the stages of the trajectory (iPSCs feeder free, iPSCs on feeder, PGCLCs, expansion, Figure 6C). The four clusters to which the reviewer refers are gene clusters that are defined by unsupervised clustering analysis of variably expressed genes across the samples (Figure 6D). As it is defined computationally, it is not possible to unequivocally define gene clusters by particular cell types. However, we found that these gene clusters revealed insightful patterns (1, genes higher in cjiPSCs; 2, genes higher in cjPGCLCs; 3, genes higher in expansion culture cjPGCLCs; 4, genes higher in d2 cjPGCLCs). We have added sample information to the Figure 6D to further clarify the meaning of the data and a brief explanation of gene clusters in the figure legend. To define the trajectory in a more unbiased manner, we performed scRNA-seq and have added additional trajectory analyses (Figure 7A-K in the revised manuscript). Moreover, we also added the transcriptomic comparison as the reviewer suggested (Figure 7L, M in the revised manuscript).

9. The potential role of the Ubiquitin-Proteasome System (UPS) should only be described in the discussion. Evidence for this is not robust enough to be reported in the results, as this is based on one GO term, for "ubiquitin-dependent protein catabolic process".

We have removed this description from the result section.

10. Results displayed from whole genome bisulphite sequencing are quite limited. A more detailed analysis of this dataset should be provided.

This bisulfite sequencing is based on shallow sequencing that allowed estimation of the overall genome-wide DNA methylation levels and was not intended for base-resolution mapping of DNA methylation. Such analyses requires labor intensive and highly expensive efforts. We believe that our approach provided sufficient information to substantiate our claim and defer the base-resolution DNA methylome analyses to our next paper, which is part of our U01 consortium specific aims.

11. The generation of DDX4 and DAZL positive cells in reconstituted testis culture that represent prospermatogonia are described as "a few". Please provide quantification to indicate the efficiency of generating these cells.

By IF analyses, we identified 0.89 and 3.3% of DAZL or DDX4 positive cells, respectively (DDX4^+^TFAP2C^+^ cells [4/123, 3.3% among all TFAP2C^+^ cells] and DAZL^+^TFAP2C^+^ cells [2/232, 0.86% among all TFAP2C^+^ cells]). Overall scarcity of cells and lack of fluorescence reporters (DAZL and DDX4 are cytoplasmic proteins necessitating technically challenging intracellular staining procedure to be assessed by flow cytometry), we were not able to provide the flow cytometric plots in this study. This point has been described in the revised manuscript (page 11, Results, “Maturation of cjPGCLCs into early prospermatogonia-like state”).

12. There appears to be a direct contradiction in the discussion, with a statement made in the results. The results state that the OF/IWR1 condition suppressed the problematic expression of genes involved in meso/endoderm differentiation. However, the discussion states that transcriptomic analysis revealed no differences in genes associated with meso/endodermal differentiation between iPSC culture conditions. Please clarify this.

This has been corrected.

13. Because IWR1 had no effect on iPSCs in FF conditions, the authors suggest factors in Pluristem better support the undifferentiated state. However, these cells are not germline competent. The best outcomes have been produced by OF conditions with IWR1. No explanation is provided for why these conditions support differentiation to the germ lineage. While this is attributed to the upregulation of genes associated with the UPS, no explanation is given for what factors in the media have caused this upregulation. An additional explanation is required here to provide a link between the growth factors and small molecules present in the superior condition, and the upregulation of these genes. Furthermore, the role of WNT inhibition in germline specification requires elucidation since it dramatically increases efficiency in the OF conditions. How might inhibition of this pathway interact with the OF conditions?

It is unclear at this point why Wnt inhibition in OF cjiPS culture conditions facilitates germ cell specification upon cjPGCLCs induction. One possibility is that some of the cells in OF cjiPSCs are already primed towards different lineages. Another possibility is that this relates to differential expression of the UPS system. Future mechanistic studies are required to resolve this issue.

14. Figure 6: D: Please provide a legend for the clusters; E and F: It is not clear which plot represents which gene.

Figure 6E and F are intended to show the overall pattern of mesodermal and endodermal genes transiently upregulated at d2 cjPGCLCs. Readers can refer to Figure 6G and Figure 6—figure supplement 2 for key gene expression. Also note that these findings were confirmed by scRNA-seq analyses (Figure 7F, G), for which some of the key genes have also been shown (Figure 7F).

References

Sasaki K, Yokobayashi S, Nakamura T, Okamoto I, Yabuta Y, Kurimoto K, Ohta H, Moritoki Y, Iwatani C, Tsuchiya H, Nakamura S, Sekiguchi K, Sakuma T, Yamamoto Takashi, Mori T, Woltjen K, Nakagawa M, Yamamoto Takuya, Takahashi K, Yamanaka S, Saitou M. 2015. Robust in vitro Induction of Human Germ Cell Fate from Pluripotent Stem Cells. Cell Stem Cell 17:178–194. doi:10.1016/j.stem.2015.06.014

Sekl Y, Yamaji M, Yabuta Y, Sano M, Shigeta M, Matsui Y, Saga Y, Tachibana M, Shinkai Y, Saitou M. 2007. Cellular dynamics associated with the genome-wide epigenetic reprogramming in migrating primordial germ cells in mice. Development 134:2627–2638. doi:10.1242/DEV.005611